# Glyco-engineered MDCK cells display preferred receptors of H3N2 influenza absent in eggs used for vaccines

Chika Kikuchi [1,2,8], Aristotelis Antonopoulos [3,8], Shengyang Wang[1,2], Tadashi Maemura[4], Rositsa Karamanska[3], Chiara Lee[3], Andrew J. Thompson[1,2], Anne Dell[3], Yoshihiro Kawaoka[4,5,6,7], Stuart M. Haslam [3] ✉ & James C. Paulson [1,2] ✉

Evolution of human H3N2 influenza viruses driven by immune selection has narrowed the receptor specificity of the hemagglutinin (HA) to a restricted subset of human-type (Neu5Acα2-6 Gal) glycan receptors that have extended poly-LacNAc (Galβ1-4GlcNAc) repeats. This altered specificity has presented challenges for hemagglutination assays, growth in laboratory hosts, and vaccine production in eggs. To assess the impact of extended glycan receptors on virus binding, infection, and growth, we have engineered N-glycan extended (NExt) cell lines by overexpressing β3-N-acetylglucosaminyltransferase 2 in MDCK, SIAT, and hCK cell lines. Of these, SIAT-NExt cells exhibit markedly increased binding of H3 HAs and susceptibility to infection by recent H3N2 virus strains, but without impacting final virus titers. Glycome analysis of these cell lines and allantoic and amniotic egg membranes provide insights into the importance of extended glycan receptors for growth of recent H3N2 viruses and relevance to their production for cell- and egg-based vaccines.

Influenza A virus has two surface glycoproteins, hemagglutinin (HA) and neuraminidase (NA), that mediate the interaction of the virus with host cell glycan receptors[1,2]. HA attaches the virus to the surface of host cells by binding to sialic acid-containing glycan receptors, while NA cleaves virus receptors by hydrolyzing sialic acid linkages and release of newly-formed virus particles, allowing infection to subsequently spread to surrounding cells[2]. Maintaining an appropriate balance of HA and NA activities against the host glycome is essential to maintain virus fitness[3,4].

These two glycoproteins are also major viral antigens and are thus subject to frequent mutation as a result of natural selection to evade host immunity[1,5]. Since its introduction in the pandemic of 1968, the

H3N2 influenza A virus has been circulating in humans for more than 50 years[6]. Despite the accumulation of numerous antigenic mutations under immune selective pressure, the virus has remained fit for transmission in the human population[7], attesting to H3N2 viruses having successfully maintained efficient binding to sialic acid-containing receptors on the human airway.

Human influenza viruses specifically recognizes glycan receptors with N-acetylneuraminic acid (NeuAc, or sialic acid) linked α2-6 to galactose (NeuAcα2-6 Gal), referred to as human-type receptors, in contrast to avian viruses that bind predominantly to α2-3-linked sialic acids (NeuAcα2-3 Gal)[8]. Over the last 25 years, however, it has become evident that mutations in the human H3 HA were increasingly

[1]Department of Molecular Medicine, The Scripps Research Institute, La Jolla, CA, USA. [2]Department of Immunology and Microbiology, The Scripps Research Institute, La Jolla, CA, USA. [3]Department of Life Sciences, Imperial College London, London SW7 2AZ, UK. [4]Influenza Research Institute, Department of Pathobiological Sciences, School of Veterinary Medicine, University of Wisconsin-Madison, Madison, WI, USA. [5]Division of Virology, Department of Microbiology and Immunology, Institute of Medical Science, The University of Tokyo, Tokyo, Japan. [6]The Research Center for Global Viral Diseases, National Center for Global Health and Medicine Research Institute, Tokyo, Japan. [7]Pandemic Preparedness, Infection and Advanced Research Center, The University of Tokyo, Tokyo, Japan. [8]These authors contributed equally: Chika Kikuchi, Aristotelis Antonopoulos. ✉e-mail: s.haslam@imperial.ac.uk; jpaulson@scripps.edu

impacting its receptor binding properties[9–12]. The first indication was a gradual loss of HA-mediated binding of viral isolates to chicken red blood cells (RBC) used in standard laboratory hemagglutination and hemagglutination inhibition assays[13–16]. Human H3N2 viruses isolated after 2000 became more difficult to grow in Madin-Darby canine kidney (MDCK) cells, the standard mammalian host cell line[10,12,17]. A series of analyses of the avidity and specificity of H3 HAs to human-type receptor glycans by biolayer-interferometry (BLI)[12] and glycan microarrays[18,19] showed H3 HAs have gradually lost their avidity to the exemplary human-type receptor Neu5Acα2-6 Galβ1-4GlcNAc, and by 2010 a majority of strains retained little or no ability to bind this model receptor fragment. Subsequent analysis using an expanded glycan library showed that H3 HAs and H3N2 viruses have evolved to bind specifically to α2-6-sialylated extended glycans, which have three or more LacNAc (Galβ1-4GlcNAc) repeats, and lost the avidity to the short glycans[20–22]. This narrowed receptor specificity could account for the reduced hemagglutination of chicken RBC and poor growth in MDCK cells, as their glycomes mostly consist of short glycans with one or two LacNAc units[23,24].

To improve the growth of human influenza viruses in MDCK cells, several genetically engineered cell lines have been developed to date[25–27]. MDCK-SIAT1 (SIAT) cells were developed by stably overexpressing an α2-6-sialyltransferase ST6GAL1 to increase the presentation of human-type Neu5Acα2-6Gal receptors on the cell surface[23,25]. SIAT cells dramatically improved the growth of strains after 2003, recovered human H3N2 clinical specimens more efficiently, and isolated viruses were shown to exhibit increased genetic stability in HA and NA genes upon serial passage in SIAT cells[10,12,17,28–30]. Takada et al[26]. further advanced this strategy by eliminating all competing α2-3-sialyltransferases in addition to stably overexpressing ST6GAL1 in MDCK cells[23,26]. The resulting humanized MDCK (hCK) cells exhibit high recovery of H3N2 viruses from clinical samples isolated in 2017–2018, and allowed the propagation of viruses to high titer with fewer mutations in HA and NA upon serial passage compared to its parental MDCK cells[26]. Thus, engineering MDCK cells to overexpress ST6GAL1 in SIAT and hCK substantially improved infection and propagation of recent H3N2 viruses[10,12,17,26,29].

These strategies, however, focused on the density of human-type α2-6-sialylated receptors without regard for the underlying poly-LacNAc structures that are essential for the receptor recognition by recent H3N2 influenza viruses. Here we have developed MDCK, SIAT, and hCK cells with N-linked glycans with poly-LacNAc extensions (NExt) by overexpressing an N-acetylglucosaminyltransferase, B3GNT2, a key enzyme for the biosynthesis of extended glycans to assess the contribution of these preferred receptors for infection and production of recent H3N2 influenza viruses. Glycoengineered cell lines, SIAT-NExt and hCK-NExt, showed higher avidity binding and single-round infection of recent H3N2 viruses without impacting final virus titers. Glycome analysis revealed that parental MDCK cells do produce some poly-LacNAc extensions but without α2-6-linked sialic acids needed for recognition by recent H3N2 viruses. In contrast, both SIAT and hCK cells have increased α2-6 sialic acids on N-linked glycans, including those with poly-LacNAc extensions, and these were further enriched in SIAT-NExt and hCK-NExt cells. Glycome analysis of allantoic and amniotic egg membranes shows a complete lack of α2-6-sialylated poly-LacNAc extended glycans accounting for poor growth of recent H3N2 viruses in eggs. The results are discussed for their implications for the production of egg- and cell-based influenza virus vaccines.

## Results

### Engineering MDCK cell lines with extended N-linked glycans

Our goal was to engineer MDCK cell lines to express N-linked glycans with poly-LacNAc extensions capped with α2-6-linked sialic acids, the preferred receptors of recent H3N2 influenza viruses (Fig. 1a). Biosynthesis of poly-LacNAc units is a coordinated reaction of two glycosyltransferases. A complex type N-linked glycan core with a single LacNAc can be further extended with a LacNAc unit by the action of a β1,3-N-acetylglucosaminyltransferase (β3GnT) followed by a β4-galactosyltransferase (β4GalT). This structure can then be further extended by the addition of another LacNAc unit by the same two enzymes or capped by a sialic acid by a sialyltransferase (Fig. 1a). Since the β4GalT is constitutively expressed in all mammalian cells, we hypothesized that the expression of β3GnTs limits the poly-LacNAc extensions, and a shift to longer extensions could be accomplished by overexpressing human beta-1,3-N-acetylglucosaminyltransferase 2 (B3GNT2), which is known for the efficient extension of N-linked glycans (Fig. 1a)[31,32].

To test this hypothesis, we transfected MDCK, SIAT, and hCK cells with a plasmid encoding the gene for human B3GNT2. Following selection and growth at limiting dilution, single clones were screened for glycan extensions by flow cytometry using A/Victoria/361/2011

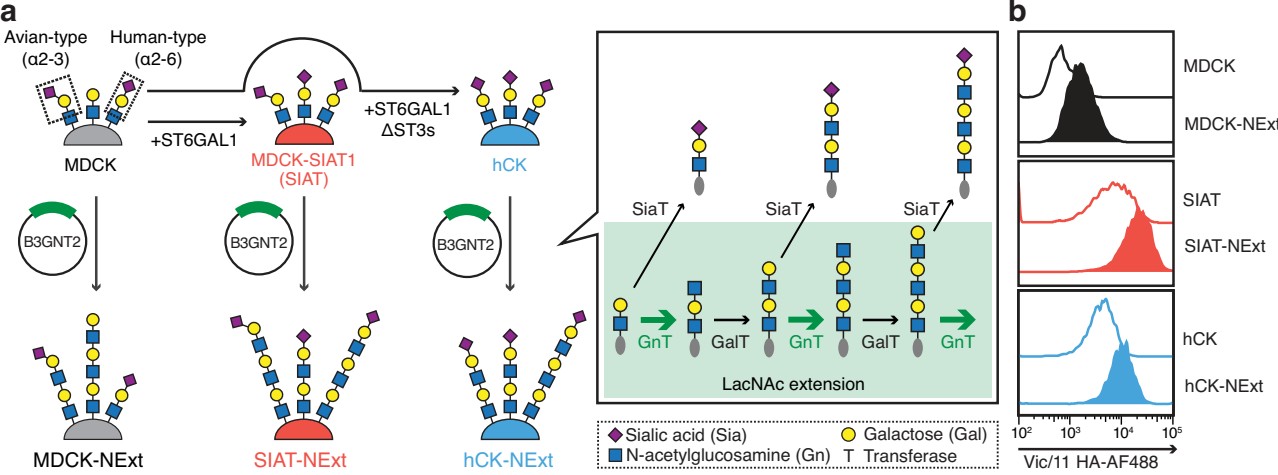

**Fig. 1 | Development of N-glycan Extended (NExt) cell lines.** MDCK, SIAT, and hCK cell lines were transfected with the β3-N-acetylglucosaminyltransferase-2 (B3GNT2) to enhance the expression of α2-6 sialylated poly-LacNAc structures recognized as preferred receptors by recent H3N2 viruses. **a** Illustration of the strategy for developing NExt cells. MDCK cells, and MDCK-derived SIAT and hCK cells that are engineered to express increased levels of glycans capped with α2-6 sialic acids, were further engineered with B3GNT2 to further extend glycan chains with poly-LacNAc repeats. **b** Representative flow cytometry diagrams comparing the binding of Vic/11 HA (Alexa Fluor 488) to B3GNT2 transfected NExt cell lines (solid color) relative to parental cell lines (solid line).

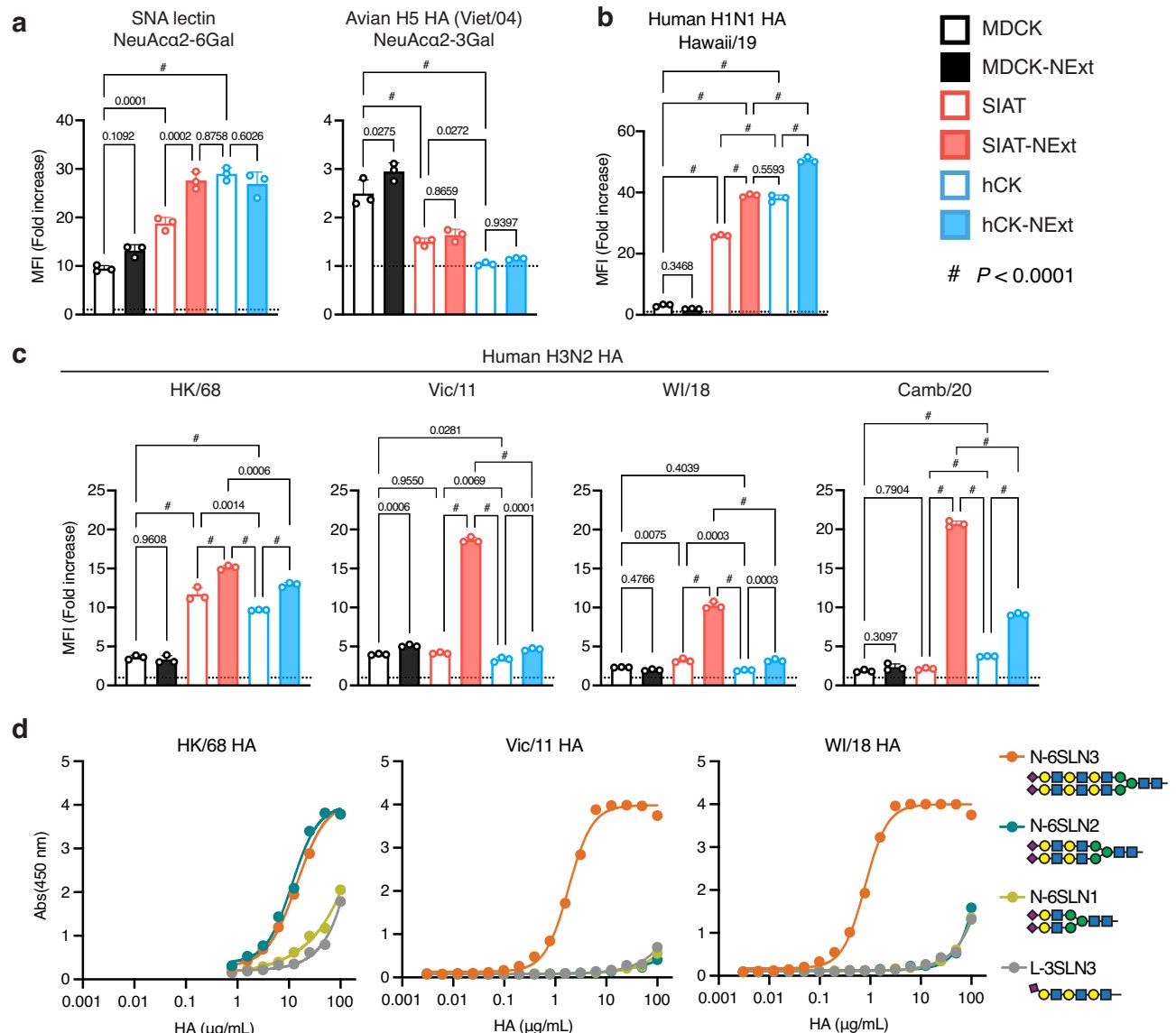

**Fig. 2 | Enhanced binding of H3 HAs from recent H3N2 isolates to SIAT-NExt cells. a–c** Flow cytometric analysis for binding of SNA-biotin or recombinant influenza virus hemagglutinins. Values represent fold increase over the negative control without SNA or rHA (dotted line). Cells were probed with (**a**) SNA lectin (NeuAcα2-6 Gal specific; left panel) or recombinant avian H5N1 Viet/04 HA (NeuAcα2-3 Gal specific; right panel), (**b**) with a recent human H1N1 (Hawaii/19) HA, or (**c**) H3 HAs from representative H3N2 isolates including HK/68, Vic/11, WI/18, and Camb/20. **d** Specificity of HK/68 (left), Vic/11 (middle), and WI/18 (right) for α2-6-

sialylated N-glycans with tri- (N-6SLN3), di- (N-6SLN2), or mono-LacNAc (N-6SLN1) repeats assessed in an ELISA format. An α2-3-sialylated tri-LacNAc linear glycan (L-3SLN3) was used as a control. **a–c** Bars indicate the mean of three replicate experiments. **d** Data indicate the mean of two replicate experiments. **a–c** Error bars indicate the standard deviation. *P* values calculated using the one-way ANOVA with Tukey's multiple comparison test are shown as actual values except for *P* < 0.0001 designated as #. **a–d** Source data are provided as Source Data file.

(Vic/11) recombinant HA (rHA) that binds only to α2-6-sialylated gly-cans with three or more LacNAc repeats[21]. Transfected clones with the highest mean fluorescence intensity (MFI) were selected as the corresponding MDCK-NExt, SIAT-NExt, and hCK-NExt cell lines (Fig. 1b and Supplementary Fig. 1).

**Characterization of NExt cell lines for binding of H3N2 HAs**
The three parental lines and their NExt counterparts were character-ized for expression of α2-3 and α2-6 sialosides by flow cytometry using *Sambucus nigra* Agglutinin (SNA), a lectin specific for NeuAcα2-6 Gal, and an H5 rHA specific for NeuAcα2-3 Gal from an avian virus A/Viet Nam/1203/2004 (H5N1; Viet/04)[33] (Fig. 2a, Supplementary Fig. 2a). As expected, MDCK and MDCK-NExt cells showed the weakest expression of NeuAcα2-6 Gal and strongest expression of NeuAcα2-3 Gal. In con-trast, SIAT, SIAT-NExt, hCK, and hCK-NExt that overexpress ST6GAL1

showed strong expression of NeuAcα2-6 Gal and corresponding decreased expression of NeuAcα2-3 Gal, which was effective to back-ground level for hCK cells, as previously described[23,26]. For the most part, there was little change in the binding of these probes between parental and NExt cells, except for a significant increase in the binding of SNA to SIAT-NExt over its parental cell line SIAT (Fig. 2a, left panel). This is possibly due to the improved efficiency of ST6GAL1 for using extended glycans as a substrate.

Next, we evaluated the binding of a panel of rHAs from a human H1N1 pdm09 strain A/Hawaii/70/2019 (Hawaii/19) (Fig. 2b, Supple-mentary Fig. 2b), and a series of H3N2 strains: A/Hong Kong/1/1968 (HK/68); Vic/11; A/Wisconsin/04/2018 (WI/18); A/Cambodia/e0826360/2020 (Camb/20) (Fig. 2c, Supplementary Fig. 2c). The Hawaii/19 H1 rHA showed strong binding to the SIAT and hCK cell lines over MDCK cells as reported previously[10,25,26], with statistically significantly increased

binding to NExt cell counterparts, likely a result of the increased expression of glycans with two or more LacNAc units known to bind to H1 HAs with higher avidity[34,35]. Notably, binding to the hCK and hCK-NExt cell lines was higher than binding to the SIAT and SIAT-NExt cell lines.

Comparison of the rHA from the pandemic (HK/68) and more recent H3N2 strains binding to the six cell lines revealed significant differences (Fig. 2c). The HK/68 HA showed stronger binding to SIAT and hCK cells over MDCK cells with a further boost in binding to both SIAT-NExt and hCK-NExt cells (Fig. 2c), as seen for the Hawaii/19 H1 rHA, reflecting the increase in Neu5Acα2-6 Gal sequence detected with SNA (Fig. 2a). In contrast, rHAs from the more recent isolates, Vic/11, WI/18, and Camb/20, showed a significantly stronger binding to SIAT-NExt cells, followed by hCK-NExt cells. Overexpression of the B3GNT2 enzyme resulted in up to 10-fold increased binding of the HAs to SIAT-NExt cells relative to SIAT cells, and up to 2.4-fold increased binding to hCK-NExt cells relative to hCK cells.

The receptor specificity of the HK/68, Vic/11, and WI/18 rHAs was confirmed by ELISA using biotinylated synthetic N-linked glycans with one to three LacNAc repeats (Fig. 2d). HK/68 rHA shows weak binding to the α2-6-sialylated glycan with one LacNAc repeat, and higher avidity binding to glycans with two and three LacNAc repeats. In contrast, the recent H3 rHAs from Vic/11 and WI/18 exhibit strict specificity for α2-6-sialylated glycans with three LacNAc repeats and bind with nearly 10-fold higher avidity than the HK/68 rHA. Taken together, the results suggest that the expression of B3GNT2 in the SIAT-NExt and hCK-NExt cells increased the expression of extended N-linked glycan receptors preferred by recent H3N2 influenza viruses.

## Glycome analysis of MDCK parental and NExt cell lines

To directly assess the cellular glycome, N-linked glycans of MDCK, SIAT, and hCK cells and their corresponding NExt cells were analyzed by matrix assisted laser desorption ionization-time of flight mass spectrometry (MALDI-TOF MS) (Fig. 3, Supplementary Figs. 3–10, Supplementary Data 1). The primary objective was to determine the degree to which glycans were capped with human-type NeuAcα2-6Gal sequence and to determine the extent to which poly-LacNAc extension is induced in NExt cells by B3GNT2. To assess NeuAcα2-6Gal content, spectra were collected for permethylated glycans of each cell line before and after enzymatic digestion with Sialidase S (Sial-S), a bacterial sialidase that removes only α2-3-linked NeuAc residues. Since treatment with Sial-S leaves glycans capped with α2-6-linked NeuAc residues, this experiment allows us to compare the relative expression of α2-6-sialylated glycans between cell lines. For selected molecular ions of high molecular weight glycans, poly-LacNAc extensions were confirmed using MALDI-TOF/TOF MS/MS fragmentation analysis. Spectral peaks were annotated for N-glycan structures based on composition, tandem mass spectrometry, and standard biosynthetic rules that assume constant Man$_3$GlcNAc$_2$ core and LacNAc extensions with NeuAc, Fuc, or Galα1-3Gal (GalαGal) modifications manually and with the assistance of the Glycoworkbench tool[36,37]. For each cell line, we have provided a list of all molecular ions corresponding to complex N-glycans (m/z 2966 and above) normalized to the sum of their relative intensities (Supplementary Data 1).

The N-glycome of MDCK and MDCK-NExt cells consist of high mannose and complex N-glycans that are mostly core-fucosylated and capped with NeuAc residues and GalαGal structures (Supplementary Figs. 3 and 4). Low levels of bisected N-glycans comprising GlcNAcβ1-4Manβ- were also seen as reported previously[23]. After Sial-S digestion, the glycome profiles of MDCK and MDCK-NExt cells yielded different glycome profiles resulting from decreased abundance of sialylated N-glycans (Supplementary Fig. 3a, b vs Fig. 3c, d; and Supplementary Fig. 4). We conclude that the majority of sialylated glycans on MDCK and MDCK-NExt cells are α2-3-linked, consistent with findings in a recent report on the glycome of MDCK cells[23].

Poly-LacNAcs were observed in high mass glycans reaching up to 12 total LacNAc units (m/z 8181). Annotation of these structures before Sial-S treatment revealed sialylated poly-LacNAc-structures in tetra-sialylated, tri-sialylated, and di-sialylated N-glycans in order of relative abundance (Supplementary Fig. 5a, b, compare blue, orange, and red shaded areas respectively). The vast majority of sialylated poly-LacNAcs were linear, ranging 1 to 4 LacNAc units (Supplementary Fig. 6, MDCK panels). Branched poly-LacNAcs (I-branched) were also detected in low abundance, as indicated from the characteristic fragment ion at m/z 2106 (Supplementary Fig. 6f, h, MDCK panels). Importantly, glycan profiles for the MDCK and MDCK-NExt cells exhibited a minimal shift to higher molecular weight glycans before or after Sial-S treatment, suggesting little impact of the expression of the B3GNT2 enzyme in MDCK-NExt cells (Supplementary Fig. 5c, d). In summary, the glycome profiles of MDCK and MDCK-NExt cells are consistent with the weak binding of H3 HAs due to poor expression of α2-6 sialic acids and minimal impact on extension of poly-LacNAc structures by expression of B3GNT2 in MDCK-NExt cells.

Glycome analysis of SIAT and SIAT-NExt cells exhibited significant differences from MDCK and MDCK-NExt cells that could be attributed to increased expression of ST6GAL1. (Fig. 3a, b, e, f and Supplementary Figs. 7 and 8). Increased sialylation was observed across the entire mass range in both SIAT and SIAT-NExt cells. For example, N-glycans in the high-mass region revealed high capping with NeuAc in SIAT and SIAT-NExt cells compared to the MDCK cells (compare Fig. 3a, b with Supplementary Fig. 5a, b; m/z 5485 to 5124, m/z 5934 to 5573, m/z 6383 to 6022 and m/z 6833 to 6472). Importantly, nearly all sialylated glycans were retained following the Sial-S treatment (Supplementary Figs. 7c, d and 8, or see red peaks in Supplementary Fig. 7a), in contrast to MDCK and MDCK-NExt cells, which lost almost all sialylated glycans after Sial-S treatment (Supplementary Figs. 3c, d and 4, or see red peaks in Supplementary Fig. 3a). These results confirm that the majority of sialic acids in SIAT and SIAT-NExt cells are α2-6-linked, induced by the ST6GAL1 overexpression.

Increases in the relative abundance of higher mass N-glycan structures were observed in SIAT-NExt cells compared to SIAT cells, consistent with additional LacNAc extensions by B3GNT2 transfection (Fig. 3a, b, e, f, i). This can readily be seen in MALDI-TOF MS spectra of SIAT and SIAT-NExt glycans in the 5000–7000 m/z range (Fig. 3a, b). Indeed, there was a clear increase in the relative abundance of tetra-sialylated N-glycans, as revealed by the shift in the distribution of these structures towards higher masses (Fig. 3a, b, see blue shaded area). This shift to higher molecular weight species is highlighted by comparing peak relative intensities normalized to the sum of the relative intensities for either native or Sial-S treated spectra (Fig. 3e, f). Relative intensities of the tetra- and tri-sialylated glycans differing in the number of LacNAc units show an increase of up to 892% in SIAT-NExt cells compared to SIAT cells (Fig. 3i). The shift to the higher molecular weight species was retained after Sial-S treatment (Fig. 3f, i), demonstrating that the increased poly-LacNAcs contain α2-6-sialylated branches. Following Sial-S treatment, however, there was a reduction of tetra-sialylated to tri-sialylated glycans, consistent with loss of α2-3 sialic acids from one branch of the tetra-sialylated glycans (Fig. 3e, f, see blue vs yellow bars). Further characterization of the poly-LacNAc sequences on selected molecular ions by MALDI-TOF/TOF MS/MS fragmentation analysis revealed that both SIAT and SIAT-NExt cells contained NeuAc-poly-LacNAcs up to six (6) LacNAc units, with the most abundant ranging from two (2) to four (4) units long (Supplementary Fig. 6 and 9). Overall, the results show that SIAT-NExt cells have increased extended N-glycans capped with α2-6-linked sialic acids, consistent with the increased binding of recent H3 HAs (Fig. 2).

N-linked glycans of hCK and hCK-NExt cells exhibited high levels of terminal α2-6-linked sialic acid similar to SIAT and SIAT-NExt cells, and they also had poly-LacNAc extensions (Fig. 3c, d, Supplementary

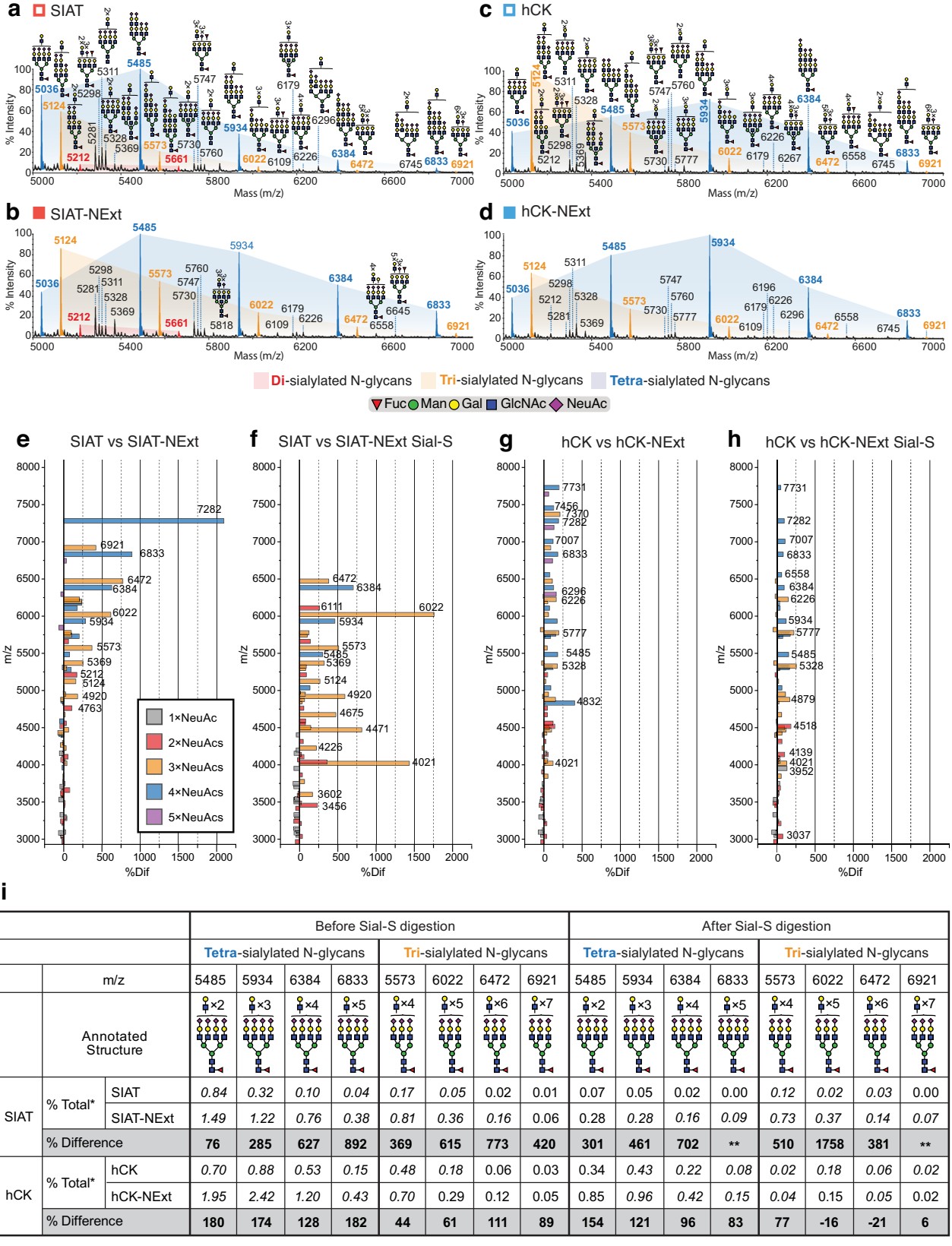

Fig. 10). Concerning the relative abundance of α2-6-linked sialic acids, a comparison of the MALDI-TOF MS spectra of hCK and hCK-NExt cells before (Supplementary Fig. 10a, b) and after (Supplementary Fig. 10c, d) treatment with Sial-S did not reveal any significant changes, in accordance with the vast majority of the sialic acids being α2-6-linked as reported previously[23,26].

As seen in the MALDI-TOF MS spectra in the m/z 5000–7000 range, there was also a high relative abundance of poly-LacNAc tetra- and tri-sialylated N-glycans (Fig. 3c, d). Notably, however, there was only a minor shift in the distribution of N-glycans of hCK-NExt towards higher masses compared to hCK cells before or after Sial-S treatment (Fig. 3g, h), an observation also reported recently for B3GNT2 transfected hCK

**Fig. 3 | Partial MALDI-TOF mass spectra of N-glycans of SIAT, SIAT-NExt, hCK, and hCK-NExt cell lines. a–d** MALDI-TOF MS analysis of permethylated N-glycans from (**a**) SIAT; (**b**) SIAT-NExt; (**c**) hCK; (**d**) hCK-NExt. Red, yellow- and blue-shaded areas highlight the distribution shift of poly-LacNAcs on bi-, tri-, and tetra-sialylated N-glycans, respectively. Shaded areas were manually inserted to assist clarity. The same colored peaks differ only in the number of LacNAcs present. All molecular ions are [M+Na]⁺. Putative structures are based on composition, tandem mass spectrometry, and biosynthetic knowledge and annotated manually with the aid of GlycoWorkBench[36]. Annotated structures were drawn according to the Symbol Nomenclature for Glycans (SNFG) guidelines[63]. Full MALDI-TOF mass spectra of the N-glycans can be found in Supplementary Fig. 7 (SIAT and SIAT-NExt cells), Supplementary Fig. 10 (hCK and hCK-NExt cells), and full annotations in Supplementary Fig. 8. **e–g** Percent difference of the molecular ions corresponding to complex N-glycans (m/z 2966 and above) calculated between (**e**) SIAT vs SIAT-NExt cells, (**f**) SIAT vs SIAT-NExt cells after Sial-S treatment, (**g**) hCK vs hCK-NExt cells, and (**h**) hCK vs hCK-NExt cells after Sial-S treatment. The percentage difference was calculated after normalizing the relative intensity of each molecular ion to the sum of their relative intensities for each of the above cell lines. The data can be found in Supplementary Data 1. **i** Relative quantification of the peak areas of tetra- and tri-sialylated N-glycans annotated in blue- and yellow-colored peaks, respectively, in (**a–d**). Italic numbers represent glycan species confirmed to contain three (3) or more linear LacNAc repeats by tandem mass spectrometry. *% Difference was calculated as described above (**e–g**). **% Difference not calculated because the molecular ions on the control were not detected. Source data are provided as Source Data file.

cells[38]. This is in contrast to the case for the SIAT-NExt cells (compare Fig. 3e–g). The relative intensities of the molecular ions in the hCK-NExt compared to hCK cells for the tetra- and tri-sialylated glycans was increased by 182%, in contrast to the 892% increase in SIAT-NExt cells (Fig. 3i). Additional MALDI-TOF/TOF MS/MS fragmentation analysis confirmed that the B3GNT2 expression in the hCK-NExt cells did not affect the distribution of poly-LacNAcs chains, with both having sialic acid terminated poly-LacNAcs with up to 5 LacNAc units (Supplementary Figs. 6, 9).

## Glycome analysis of chicken egg amniotic and allantoic membranes

Given the importance of chicken eggs for the growth of human influenza viruses and vaccine production and the difficulty of growth of H3N2 influenza viruses in eggs without selection of receptor binding variants, we also analyzed the N-linked glycans from membranes lining the allantoic and amniotic cavities of the chicken embryo (Fig. 4, and Supplementary Fig. 11, allantoic; Supplementary Fig. 12, amniotic). While influenza viruses are grown in the allantoic cavity for large scale production of vaccines, human influenza viruses have historically been shown to be more easily recovered in the amniotic cavity[39,40]. Previous reports found that the major complex type N-linked glycans of the amniotic and allantoic membranes were predominantly di- and tri-branched glycans with a single LacNAc unit capped with α2-3- and α2-6-linked sialic acids[41]. For direct comparison of the two egg membrane glycomes to the MDCK cell lines, we performed MALDI-TOF MS and TOF/TOF MS/MS analysis before (Fig. 4a, Supplementary Fig. 12a) or after (Supplementary Figs. 11 and 12b) enzymatic digestion with sialidase A (Sial-A). In addition to high-mannose type glycans found in all cells, complex N-glycans were observed in both allantoic and amniotic membranes ranging from bi- to penta-antennary structures, mainly being core-fucosylated and capped with NeuAc residues. In contrast to the MDCK, SIAT, and hCK cell lines, NeuAc-LacNAc extensions were of very minor relative abundance, and when found, they were principally restricted to two (2) LacNAc repeats. Sial-S digestion of allantoic and amniotic cell membrane-derived N-glycans to remove α2-3-linked sialic acids revealed sialic acid-containing glycans with α2-6-linked NeuAc residues in both types of membranes (Fig. 4b and Supplementary Fig. 12c). Based on the proportion of sialic acids removed after sialidase S digestion, α2-3-linked sialic acids are more abundant than α2-6-linked sialic acids in both membranes, a conclusion reached in previous studies[41]. Taken together, the data indicate that N-glycans from amniotic and allantoic chicken cell membranes contained minimal LacNAc units that were limited principally to two (2) LacNAc repeats long capped mainly with α2-3-linked and less with α2-6-linked sialic acids.

## Susceptibility of MDCK cell lines to infection and propagation of H3N2 viruses

To determine if the increased avidity of H3 HAs for binding to SIAT-NExt cells (Fig. 2) impacts their susceptibility to virus infection, we first employed a single-round infectivity assay with a short adsorption time to focus the assay on the efficiency of the adsorption of virus to surface receptors. Cells were inoculated with the virus for 10 min to allow viral adsorption and then washed to remove non-adsorbed viruses. They were then incubated for 16 h in a fresh medium without tosyl-phenylalanyl-chloromethyl ketone (TPCK)-trypsin to limit the infection to a single-round of replication. This assay enables us to compare cells for the efficiency of the HA-mediated adsorption step in virus infection.

Representative H3N2 virus strains tested for HA binding (Fig. 2), HK/68, Vic/11, and WI/18 were evaluated for infection of the six MDCK cell lines in the single-round infectivity assay (Fig. 5a). The introductory HK/68 pandemic strain infected all six cell lines with similar efficiency as expected based on binding of the recombinant HA to these cells. For the more recent strains Vic/11 and WI/18 with extended glycan specificity, SIAT-NExt cells showed higher susceptibility over other cell lines. The difference was the most remarkable for WI/18, which showed significantly higher infectivity to SIAT-NExt over the other five cell lines. These results are consistent with the HA avidity assays in which Vic/11 and WI/18 showed the highest avidity binding to SIAT-NExt cells (Fig. 2c).

To confirm that this observation applies to the clinical isolates, we ran the single-round infectivity assays against H3N2 virus isolates from 2017–2018 season, A/Tokyo/UT-DA23-1/2017(DA23/17), A/Tokyo/UT-DA30/2018 (DA30/18), A/Tokyo/UT-HP62/2018 (HP62/18), which were previously shown to efficiently infect hCK cells over MDCK cells[26], and two isolates from 2021, A/Yamagata/1/2021 (Yamagata/21), and A/Miyagi/1/2021 (Miyagi/21) (Fig. 5b). All virus isolates showed highest infectivity to SIAT-NExt cells compared to other three cell lines tested, consistent with higher avidity binding of H3 HAs of other recent H3N2 influenza viruses to extended receptors on SIAT-NExt cells (Fig. 2c).

Six recent human H3N2 influenza isolates (DA23/17; DA30/18; HP62/18; A/Yokohama/1/2021; A/Yokohama/2/2021; and A/Yokohama/3/2021) were also tested for growth kinetics to see how viral adsorption translated to virus propagation (Fig. 5c). Overall, while these recent H3N2 viruses grew poorly in MDCK and MDCK-NExt cells, they grew efficiently in SIAT, SIAT-NExt, hCK, and hCK-NExt cells with no significant difference in viral titers at 72 h post-infection. Interestingly, the growth in SIAT-NExt cells was delayed for several viruses, showing lower titers at 24 h compared to the other three cell lines (Fig. 5c). One possible explanation is that the strong avidity of the viruses for SIAT-NExt cells hindered their diffusion from infected cells at early rounds of infection. Notably, this delay in kinetics coincides with a reduced plaque size of H3N2 viruses grown in SIAT-NExt cells compared to hCK-NExt cells, as shown for WI/18 in Fig. 5d. Collectively, these results showed that SIAT, SIAT-NExt, hCK, and hCK-NExt cells are equally efficient in multi-round replication of recent H3N2 viruses.

## Discussion

Currently circulating H3N2 viruses, dominated by subclade 3C2.a, exhibit strict specificity for α2-6-sialylated extended glycans with three or more LacNAc repeats, having gradually lost avidity towards short

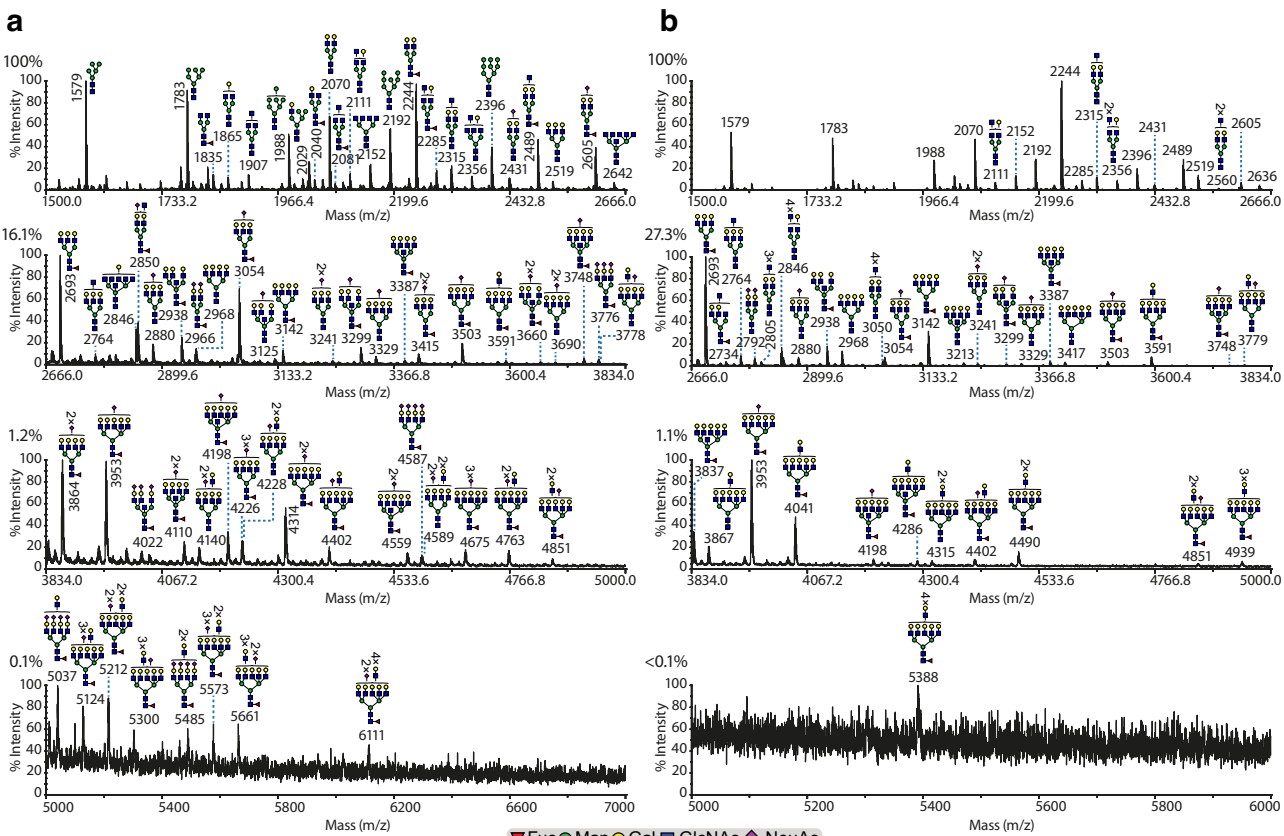

**Fig. 4 | MALDI-TOF MS analysis of permethylated N-glycans from allantoic membrane before and after Sial-S digestion.** MALDI-TOF MS analysis of permethylated N-glycans from allantoic membrane (**a**) before and (**b**) after Sial-S digestion. Putative structures are based on composition, tandem mass spectrometry, and biosynthetic knowledge and annotated manually with the aid of GlycoWorkBench[36]. Percentages on the top left of each panel correspond to the relative percentage of the maximum peak of the corresponding panel compared to the relative intensity of the maximum peak of the top panel. N-glycans with core-

fucosylation had higher relative abundance when compared to their non-core-fucosylated counterparts (compare m/z 2850 vs 2676, 3054 vs 2880, 3299 vs 3125, 3503 vs 3329). Compare panels a and b for the decrease of the relative abundance of the α2-3-linked NeuAc residues. Similarly, compare the relative abundance of the tetra-antennary N-glycans with various levels of NeuAc residues (m/z 3503, 3864, 4314, and 4587) compared to the base peak tetra-antennary N-glycan without any NeuAc residue (m/z 3142). Source data are provided as Source Data file.

glycans over the last two decades[11,20–22,42]. This change in receptor recognition presents challenges for growing viruses in MDCK cells or eggs without selection of hemagglutinin (or neuraminidase) variants that can also alter antigenicity[26,28,29,43–48]. In this report, given the importance of MDCK cells in laboratory isolation and cell-based production of influenza viruses, we engineered MDCK cells, and MDCK-derived SIAT and hCK cells, to overexpress an extension enzyme, B3GNT2, to enhance N-glycan extension (NExt) and assessed its impact on H3N2 virus HA binding and virus infection.

We found that SIAT-NExt cells, and to a lesser extent hCK-NExt cells, exhibit increased HA binding to the cell surface and improved susceptibility to infection by recent H3N2 virus strains compared to MDCK cells (Figs. 2 and 5). Glycome analysis showed that MDCK cells have the capacity to produce extended N-linked glycans as recently reported[23], however, they exhibit poor binding of HA and growth of H3N2 viruses because they lack sufficient capping with α2-6-linked sialic acid. By contrast, glycans on SIAT and hCK cells are predominantly capped with α2-6-linked sialic acids as a result of over-expression of ST6GAL1[23]. Glycome profiles of SIAT-NExt cells compared to that of parental SIAT cells show a substantial shift towards higher molecular weight glycans, and a similar but weaker shift was shown for hCK-NExt cells compared to hCK cells (Fig. 3). Collectively, the glycome profiles support the importance of α2-6-sialylated, poly-LacNAc extended N-glycans in the high-avidity binding and infection of these cells by recent H3N2 viruses.

It is evident that SIAT-NExt cells exhibited the highest avidity binding of the HA of recent H3N2 viruses, and H3N2 virus infection in the single round infection assay, showing these cells have more functional glycan receptors on the surface (Figs. 2 and 5). Yet the hCK and hCK-NExt cells exhibited similar high molecular weight glycan profiles. There is a limit to which the glycome analysis can provide information for a structural explanation for these differences, particularly when the shift to higher masses represents a small percentage of the total glycome. While the number of LacNAc units in a large glycan can be determined by molecular weight and poly-LacNAc extensions of multi-antennary glycans can be derived by MALDI-TOF MS/MS analysis, our analysis indicates multiple structural isomers for each molecular ion. For example, a tetra sialylated glycan with 8 LacNAc units can theoretically make tetra-antennary glycans each branch containing only 2 LacNAcs or with one branch extended with three or more LacNAcs, and the other branches with only one or two LacNAcs. It is also not possible to accurately derive absolute quantitative levels of N-glycans between MALDI-TOF MS spectra. Such limitations could obscure meaningful structural differences in quantitative comparisons of the glycomes between cell lines.

It is likely there is variation in poly-LacNAc extension of glycans in SIAT-NExt and hCK-NExt cells that result from differential expression of other glycosyltransferases that compete with the sialyltransferases and B3GNT2 enzymes during biosynthesis. One example of such competitive glycosylation is evident in the glycan profiles of MDCK

 

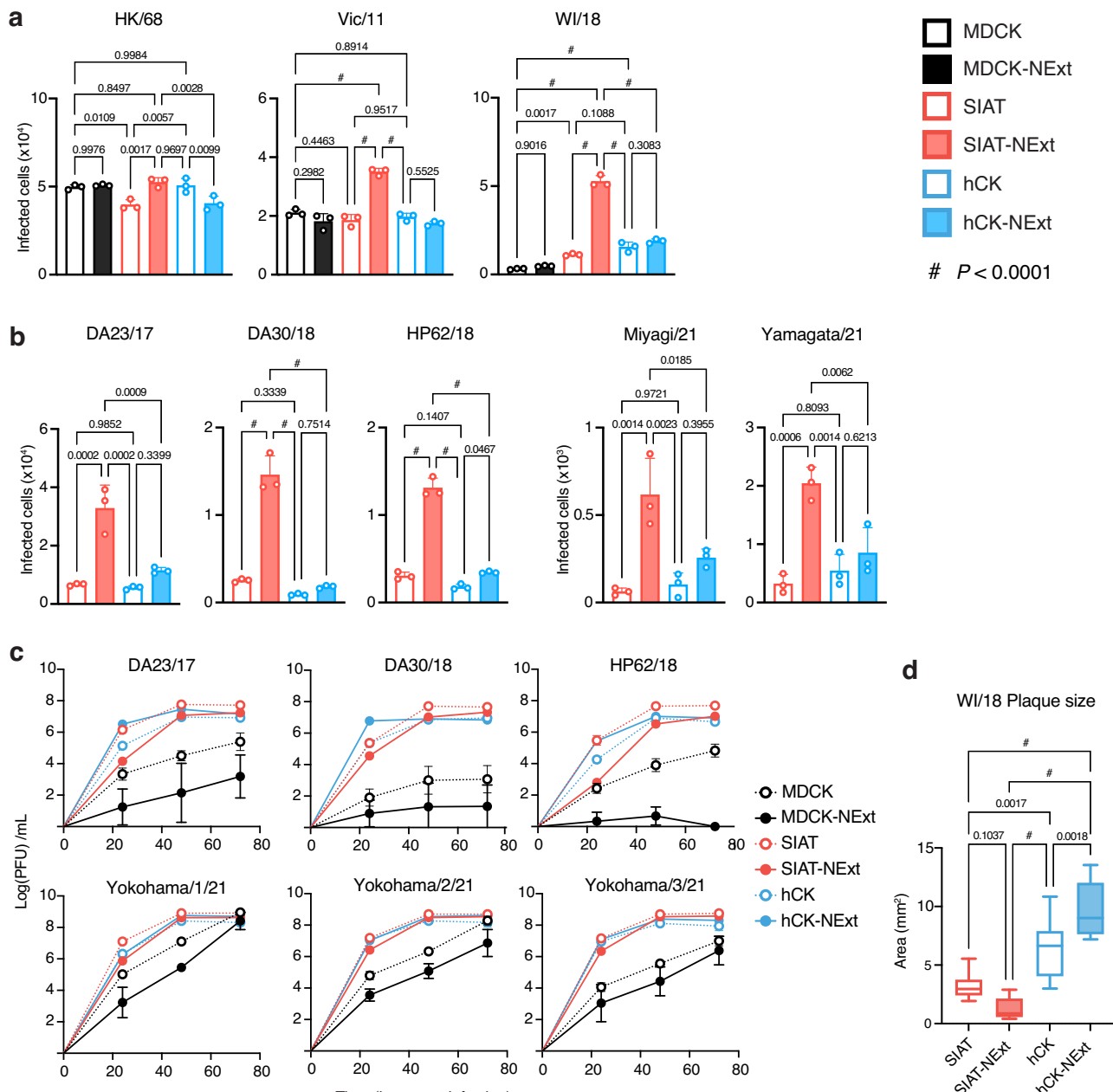

**Fig. 5 | Susceptibility of cell lines to infection by recent H3N2 viruses.** Representative strains and recent clinical isolates of the human H3N2 virus were evaluated for infection of MDCK, SIAT, hCK, and corresponding NExt cell lines. **a, b** Single-round infectivity assay to assess HA-mediated adsorption-dependent virus infectivity. **c** Growth kinetics of viruses in the cell lines. **d** Quantification of plaque size of WI/18 virus in SIAT, SIAT-NExt, hCK, and hCK-NExt cell lines. 10 plaques from each cell line were used to obtain the statistical analysis. **a, b** Bars indicate the mean of three replicate experiments. **c** Points indicate the mean of three replicate experiments. **a–c** Error bars indicate the standard deviation. **d** The center line indicates the median, upper and lower box lines show quartiles; and whiskers show the maximum and minimum values. **a, b, d** $P$ values calculated using the one-way ANOVA with Tukey's multiple comparison test are shown as actual values except for $P < 0.0001$ designated as #. **a–d** Source data are provided as Source Data file.

and SIAT cells. The increased expression of ST6GAL1 in SIAT cells very effectively competes with the endogenous α2-3-sialyltransferases for a common LacNAc acceptor, resulting in a switch from capping the glycan with α2-3-linked to α2-6-linked sialic acids. Similarly, the β3GnT extension enzyme(s) also compete with sialyltransferases for the same LacNAc acceptor substrate. There is a major difference between SIAT-NExt and hCK-NExt cells in this regard. While both are engineered to express ST6GAL1, the hCK cells were further engineered to delete all the α2-3-sialyltransferases. As one consequence of the deletion of these enzymes, Byrd-Leotis *et al.* noted an increase in 3-O-sulfate-Gal in hCK cells relative to MDCK cells, which they attributed to increased

activity of the endogenous sulfotransferases in the absence of competition with α2-3-sialyltransferases[23]. While this increased 3-O-sulfate-Gal in hCK cells did not affect the high α2-6-sialylation of hCK and hCK-NExt cells as shown here, this observation suggests that deletion of the sialyltransferses induced unintended changes in the glycome. Furthermore, the observed shift to higher molecular weight glycans in hCK cells compared to MDCK cells (Fig. 3c vs Supplementary Fig. 5a) could also be attributed to the deletion of α2-3-sialyltransferases. Sialic acid deficiency is known to promote N-glycan branching[49], therefore loss of α2-3-sialyltransferases mapped to the medial- and trans-Golgi[50,51] could also promote increased branching before the

addition of sialic acid by ST6Gal I that predominately occurs at a later stage of biosynthetic pathway in the trans-Golgi and trans-reticular network[52]. Similarly, the delay of medial-Golgi capping of glycan chains in hCK cells could also promote the extension of LacNAc repeats due to less competition for the endogenous β3GnT(s) to use the LacNAc substrate for extension.

The results provide insights into the impact of increased α2-6-sialylated extended receptors for adsorption to and infection by recent H3N2 influenza viruses. Clearly, the increased receptors on SIAT-NExt cells permit more avid adsorption of the HA (Fig. 2). As a result, in an assay favoring adsorption as a limiting step, there is a more efficient infection of the cells (Fig. 5a, b). For some H3N2 strains, however, slower kinetics of virus production were observed as a consequence of higher avidity. The higher avidity of HA can result in in viruses being more slowly released from the infected cells by the NA, resulting in reduced plaque size and slower kinetics of virus production, without impacting the final production of the virus (Fig. 5c). It has been clearly demonstrated that the expression of ST6GAL1 in SIAT and hCK cells has improved recovery of human H3N2 viruses from clinical isolates compared to parental MDCK cells[26,53]. The even higher avidity binding of H3N2 viruses to SIAT-NExt cells indicates that it may also be of value in the isolation of new H3N2 influenza viruses from clinical samples.

We believe that these results are particularly relevant to the production of recent H3N2 viruses for inactivated influenza viruses for egg-based and cell-based vaccines. Changes in receptor specificity upon adaptation of human influenza viruses to eggs have been observed since the early days of Burnet[11,54], and it is now well documented that recent H3N2 isolates grow poorly in eggs and acquire adaptive mutations that exhibit altered receptor binding properties[44–47]. The viruses for egg-based vaccines are produced by cells in the membranes lining the allantoic cavity[55]. Glycome analysis reported here and by others clearly show that allantoic membranes have N-glycans with only one or possibly two LacNAc repeats[41] representing a mismatch for the receptor specificity of recent H3N2 viruses that require extended glycans. This glycan-mismatch will further drive the selection of mutants that alter receptor binding for improved growth, and as a result, can change antigenicity that impacts vaccine efficacy[45–47,56,57]. Currently, MDCK cells are used for the production of cell-based vaccines given the viruses grown in the MDCK cells show fewer mutations than egg-grown viruses[45,48,58]. However, given the poor growth of recent H3N2 viruses in MDCK cells, the potential for selection of receptor binding variants is still a concern[9]. Indeed, extended glycans of MDCK cells are mostly α2-3-sialylated, therefore are sub-optimal for the growth of recent H3N2 viruses (Supplementary Figs. 3–5). In contrast, we have shown that SIAT and hCK cells have sufficient amounts of the α2-6-sialylated extended glycan receptors to support the production of viruses equivalent to their NExt counterparts that display even more functional receptors (Figs. 3, 5c). In this regard, several groups have reported that viruses maintain higher genetic stability when grown on either SIAT or hCK cells compared to MDCK cells[17,26,28,29]. Thus, there is strong mechanistic justification to consider the use of hCK or SIAT cells for the production of H3N2 viruses for cell-based vaccines.

## Methods
### Ethical statement
The experiments using live seasonal human influenza were approved by the Scripps Research Institutional Biosafety Committee under IBC protocol 04-08-20-01 "Influenza/HIV/Coronavirus".

### Cell culture
MDCK cells (ATCC) were maintained in MEM supplemented with 10% fetal bovine serum (FBS), 2 mM L-Glutamine, and 100 U mL⁻¹ of Penicillin-Streptomycin. MDCK-SIAT1 cells (Sigma-Aldrich) were maintained in MEM supplemented with 10% FBS, 2 mM L-Glutamine, 100 U mL⁻¹ of Penicillin-Streptomycin, and 1 mg mL⁻¹ G418 sulfate. hCK

cells (from Kawaoka lab[26]) were maintained in MEM supplemented with 5% newborn calf serum, 10 μg mL⁻¹ blasticidin, and 2 μg mL⁻¹ puromycin. Cells were incubated at 37 °C in 5% $CO_2$ unless otherwise described.

### Expression and purification of recombinant HA
Recombinant HA were expressed using previously published plasmids (A/Hong Kong/1/68, A/Victoria/361/2011[21], A/Viet Nam/1203/2004[33]) and newly constructed plasmids. HA coding sequences of A/Wisconsin/4/2018, A/Cambodia/e0826360/2020, and A/Hawaii/70/2019 were synthesized from the amino acid sequences deposited to GISAID (Isolate ID EPI_ISL_296103, EPI_ISL_296103, and EPI_ISL_397028, respectively). HA ectodomain (residues 11-521; H3 numbering) were amplified and cloned under CMV promoter on a customized DNA vector for expression in mammalian tissue culture with an N-terminal CD5 signal peptide for secretion, a C-terminal leucine zipper (GCN4) motif for trimerization following a short linker, and His8-tag in the C-terminal of the construct using the NEBuilder HiFi DNA Assembly Master Mix (New England Biolabs). Final expression constructs were transfected into HEK293F cells (Freestyle 293F, Thermo Scientific) using linear PEI (polyethylimine) at 5:1 (w/w). Typically 150 mL of $1 \times 10^6$ cells/mL culture was transfected with 150 μg plasmid with 600 μg PEI. After 6 days, cell culture supernatant was collected, and proteins were purified by IMAC using 1 mL HisTrapFF column. HAs were eluted in a gradient of Tris-HCl buffer containing 0.5 M (final) imidazole, washed, and concentrated typically to 800 μg/mL for experiments. HAs were stored at 4 °C until used for experiments and experiments were done within 5 days post-purification of the protein.

### HA/lectin binding assay
A monolayer of cultured cells were detached and digested to single cells by TrypLE Express Enzyme (Gibco). Cells were washed with PBS and stained with Zombie NIR dye (BioLegend) for Live/Dead staining. After washing with PBS three times, cells were blocked with canine Fc block (1:20, Invitrogen) on ice for 10 min. The cells were subsequently incubated with respective staining reagents on ice for 45 min: biotinylated SNA (1:500, Vector Labratories) precomplexed with Streptavidin-AF488 (1:500, Biolegend); recombinant 50 μg/mL HAs precomplexed with anti-His mouse IgG2a (25 μg/mL, J095G46, Biolegend) and Alexa fluor 488-conjugated anti-mouse IgG2a (12.5 μg/mL, RMG2a-62, Biolegend). The cells were washed twice with PBS before flow cytometry. See Supplementary Fig. 13a for the gating strategy. The data were analyzed using FlowJo software. For establishment of B3GNT2 stable overexpression (NExt) cell lines, the mean fluorescent intensity (MFI) of B3GNT2 transfectants were compared to their respective parental cell lines to assess the increase of the rHA binding by the B3GNT2 transfection. For the lectin binding and HA-avidity assays, the MFI was normalized to the negative control lacking the primary reagent (lectin or HA) to account for the differences of background MFI of each cell lines and enable the comparison across cell lines.

### Establishment of B3GNT2 stable overexpression (NExt) cell lines
Full-length human B3GNT2 gene was cloned under CMV promoter on a customized DNA vector encoding neomycin (MDCK and hCK) or hygromycin (SIAT) resistance using the NEBuilder HiFi DNA Assembly Master Mix (New England Biolabs). Electroporation was conducted using the BTX ECM600 Electro Cell Manipulator according to the manufacturer's instructions. Briefly, $2.5 \times 10^6$ MDCK, SIAT, and hCK cells were resuspended in 500 μL of PBS and mixed with 5 μg expression vector. The mixtures were transferred to 2 mm gap cuvettes and immediately electroporated at 270 V, 1040 μF capacitance, and 720 Ω resistance. After electroporation, cells were cultured with either 0.5 mg mL⁻¹ G418 sulfate (MDCK and hCK) or 25 μg mL⁻¹ hygromycin (SIAT) for selection and maintenance in addition to respective basal

media. Clones were isolated by limiting dilution and phenotypically screened by binding of Vic/11 recombinant HA which has known specificity for extended glycans to select respective NExt cell lines.

## Preparation of biotinylated glycans

Biotinylated glycans were prepared as previously described[21]. Briefly, the poly-LacNAc chains were obtained by the enzymatic extension of glycan cores with *H. pylori* β1,3-GlcNAcT and recombinant *N. meningitidi*s β4galT-galE fusion. Subsequent sialylation of the poly-LacNAc chains was achieved by using either recombinant human ST6GAL1 or rat ST3GAL3. The resultant sialosides were treated with NHS-LCLC-biotin (#21343, Thermo scientific) in the presence of DIPEA (#496219, Sigma-Aldrich) to provide the desired biotinylated linear or N-linked sialosides. Synthesized compounds were confirmed by high-resolution mass spectrometer (HRMS) as follows: 6SLN1-N, ESI TOF-HRMS m/z calculated for $C_{110}H_{183}N_{12}O_{68}S$, $[M+3H]^{3+}$: 931.0328, found 931.0348; 6SLN2-N, ESI TOF-HRMS m/z calculated for $C_{138}H_{229}N_{14}O_{88}S$, $[M+3H]^{3+}$: 1174.4543, found 1174.4567; 6SLN3-N, ESI TOF-HRMS m/z calculated for $C_{166}H_{276}N_{16}O_{108}S$, $[M+3H]^{3+}$: 1418.2117, found 1418.2285; 3SLN3-L, ESI TOF-HRMS m/z calculated for $C_{77}H_{131}N_{9}O_{43}S$, $[M+2H]^{2+}$: 1901.8061, found 1901.7952.

## ELISA with biotinylated glycans

Streptavidin-coated high binding capacity 384-well plates (Pierce) were rinsed with PBS and each well was incubated with 50 μL of a 2.4 μM solution of biotinylated glycans (6SLN1-N, 6SLN2-N, 6SLN3-N, 3SLN3-L) in PBS overnight at 4 °C. The plate was washed with PBS-T (0.05% Tween 20 in PBS) to remove the excess glycan, and each well was incubated with 100 μL of blocking buffer (1% BSA and 0.6 μM desthiobiotin in PBS) at room temperature for 1 h. The plate was subsequently washed with PBS-T (0.05% Tween 20 in PBS) and used without further processing. Purified His-tagged HA trimers were precomplexed with anti-His mouse IgG2a (J095G46, Biolegend) and HRP-conjugated anti-mouse IgG (H + L) (Thermo Fisher Scientific) in the ratio 4:2:1 (w/w/w) with 1% BSA and incubated on ice for 20 min. The highest HA concentration was typically 100 μg/mL and then diluted in twofold steps. 50 μL of precomplexed HA was added to each glycan-coated well and incubated at RT for 2 h. The wells were washed with PBS-T completely and 50 μL of TMB substrate was added. The plate was incubated at room temperature for development and then quenched by adding 50 μL of 2 M sulfuric acid. The absorbance at 450 nm was measured by BioTek Synergy H1 microplate reader and data was plotted using GraphPad Prism software.

## Processing of MDCK cell lines and egg membranes to obtain N-linked glycans

$10^7$ cells per each cell line were collected. Allantoic and amniotic membranes were extracted from 10-day old chicken embryos. All samples were treated as described previously[37,59]. Briefly, each cell line was subjected to sonication in the presence of detergent 3-[(3-Cholamidopropyl)dimethylammonio]-1-propanesulfonate hydrate (CHAPS, #10810118001, Roche), reduction in 4 M guanidine-HCl (#24115, Thermo), carboxymethylation, and trypsin (#T0303, Sigma) digestion. The digested glycoproteins were then purified by plus short HLB-Sep-Pak (#186000132, Waters Corp.). N-glycans were released by peptide N-glycosidase F (E.C. 3.5.1.52; #11365177001, Roche) digestion. Released N-glycans were permethylated using the sodium hydroxide procedure and purified by classic short $C_{18}$-Sep-Pak (#WAT051910, Waters). Permethylated N-glycans were eluted at the 50% acetonitrile fraction.

## Sialidase digestions

Cleavage of α2-3-linked sialic acids from each cell line, or α2-3- and α2-6-linked sialic acids from egg amniotic and allantoic membranes, were performed with sialidase-S (*Streptococcus pneumoniae*; E.C. 3.2.1.18;

Agilent Technologies, GK80021) and sialidase-A (*Arthrobacter ureafaciens*, E.C. 3.2.1.18; Agilent Technologies, GK80040) digestions respectively as previously described[59]. Released N-glycans by peptide N-glycosidase F were incubated in 200 μl of 50 mM sodium acetate (37 °C, pH 5.5) and 170 milliunits of the enzyme were added to the sample for 24 h. The digested N-glycans were lyophilized and purified on a classic short C18 Sep-Pak (#WAT051910, Waters), followed by the same permethylation and purification procedures as described above.

## Mass spectrometry and analysis

MS and MS/MS data were acquired a 4800 MALDI-TOF/TOF (Applied Biosystems, Darmstadt, Germany) mass spectrometer. Permethylated glycans were dissolved in 10 μl of methanol, and 1 μl of dissolved sample was premixed with 1 μl of matrix (10 mg/ml 3,4-diaminobenzophenone (#184800250, Acros Organics) in 75% (v/v) aqueous acetonitrile), spotted onto a target plate, and dried under vacuum. For the MS/MS studies the collision energy was set to 1 kV, and argon was used as collision gas. The 4700 Calibration standard kit, calmix (Applied Biosystems), was used as the external calibrant for the MS mode, and [Glu1] fibrinopeptide B human (Sigma) was used as an external calibrant for the MS/MS mode.

The MS and MS/MS data were processed using Data Explorer 4.9 Software (Applied Biosystems). The processed spectra were subjected to manual assignment and annotation with the aid of a glycobioinformatics tool, GlycoWorkBench[36]. The proposed assignments for the selected peaks were based on $^{12}C$ isotopic composition together with knowledge of the biosynthetic pathways. The proposed structures were then confirmed by data obtained from MS/MS analysis. The bar chart graph was prepared with Originlab with data prepared in Microsoft Excel with relative abundances deriving from the Data Explorer software. All figures were prepared/finalised in Adobe Illustrator.

## Preparation of H3N2 influenza viruses

The H3N2 viruses A/Hong Kong/1/68, A/Victoria/361/2011, A/Wisconsin/4/2018 were obtained from the Centers for Disease Control and Prevention Influenza Reagent Resource (CDC-IRR). They were grown in SIAT cells. Recent clinical isolates (A/Tokyo/UT-DA23-1/2017, A/Tokyo/UT-DA30/2018, A/Tokyo/UT-HP62/2018, A/Yamagata/1/2021, A/Miyagi/1/2021, A/Yokohama/1/2021, A/Yokohama/2/2021, A/Yokohama/3/2021) were provided by Kawaoka lab. They were grown in hCK cells.

SIAT or hCK cells were cultured to 90% confluence in MEM medium supplemented with 2 mM L-Glutamine and 100 U mL$^{-1}$ of Penicillin-Streptomycin. Cell cultures were washed twice in warm PBS prior to the addition of virus, typically diluted 1:1000 in MEM containing 0.1% BSA. Diluted virus was incubated with cell cultures for 1 h before being removed and replaced with growth medium supplemented with 1 μg mL$^{-1}$ tosyl phenylalanyl chloromethyl ketone (TPCK)-trypsin and further incubated at 33 °C. At day 3, culture supernatant was collected and centrifuged at 1000 × g to remove cell debris and stored at −80 °C. For titering the virus stocks, $0.75 \times 10^4$ hCK cells per well were plated in 96-well plates and incubated at 37 °C overnight to allow cells to attach. Cells were washed twice with PBS, then 100 μL of virus stock (10-fold dilution series in MEM containing 0.1% BSA) were added to the wells, and incubated at 33 °C. After 1 h, 100 μL 1.2% (w/w) colloidal microcrystalline cellulose overlay in MEM containing 0.1% BSA and 5 μg mL$^{-1}$ tosyl phenylalanyl chloromethyl ketone (TPCK)-trypsin was added and further incubated at 33 °C. After 44–48 h, the overlay was removed by suction and cells were fixed with 4% paraformaldehyde solution in PBS for 30 min. Cells were washed with PBS once then stained with anti-NP antibody (1 μg/mL, H16-L10-4R5 (HB-65), BioXcell) in PBS containing 0.1% BSA and 0.1% saponin for 2 h at RT. Cells were washed with PBS with 0.05% Tween-20 three times before stained with HRP-conjugated goat anti-mouse IgG antibody (1:2000, Southern Biotech) for 1 h at RT. After washing three times with PBS with 0.05% Tween-20, cells were incubated in KPL TrueBlue

(Seracare) to visualize the plaques. Stained plates were washed with tap water to stop the reaction and dried until plaques were counted.

## Single-round infectivity assay

Protocol was modified from the viral titering assay previously described[60]. Briefly, $10^5$ cells per well were plated in 12-well plates in the corresponding medium to allow cells to adhere to the plate, at least 6 h followed by washing with PBS. Viruses in 100 μl MEM containing 0.1% BSA ($\leq 1 \times 10^4$ pfu/mL in hCK cells) were overlayed onto cells at 33 °C to allow for viral attachment. After 10 min, virus containing medium was removed and cells were washed with PBS before adding fresh MEM containing 0.1% BSA but not TPCK-trypsin to restrain the infection to a single-round. Cells were incubated for 16 h at 33 °C to allow viral growth. After 16 h, cells did not show cytopathic effect. Cells were collected, fixed, and permeabilized with the FOXP3 Fix/Perm Buffer Set (BioLegend) following the manufacturer's protocol, and stained with FITC-conjugated anti-NP monoclonal antibody (1:100, D67J, Thermo-Fisher) for 45 min on ice. Cells were washed twice with perm buffer before being resuspended in PBS containing 1% BSA for flow cytometry. See Supplementary Fig. 13b for the gating strategy. The ratio of NP-positive population was calculated by the number of infected cells using Flowjo software and Microsoft Excel. The inoculating virus concentration was titrated to ensure the ratio of NP-positive population does not exceed 50% of the samples in any cell lines tested.

## Viral kinetics assay

MDCK, MDCK-NExt, SIAT, SIAT-NExt, hCK, and hCK-NExt cells were infected with viruses at a multiplicity of infection of 0.002, which was determined using a separate measurement of virus titer by plaque assay using hCK cells. The supernatants of the infected cells were harvested at 24, 48, 72 h post-infection, and virus titers were determined by means of plaque assays in hCK cells.

## Quantitation of plaque size

A monolayer of cells in 6-well plates were infected with WI/18 virus at the dilution that gives 10-20 plaques per well for each cell line. Cells were washed with PBS at 1 h post-infection. Cells were covered with an overlay consisting of MEM, 1.2% (w/w) colloidal microcrystalline cellulose, 2 mg mL$^{-1}$ L-Glutamine, 1 μg mL$^{-1}$ TPCK-trypsin and incubated for 3 days at 37 °C. After being fixed by 4% PFA and permeabilized with 0.1% Triton-X100, infected cells were stained with anti-NP antibody (1 μg/mL, H16-L10-4R5 (HB-65), BioXcell) and subsequently stained with a secondary HRP-conjugated goat anti-mouse IgG antibody (1:2000, Southern Biotech). Plaques were visualized with KPL TrueBlue (Seracare). Images were acquired with Keyence BZX700 Widefield fluorescence microscope and analyzed using image analysis software ilastik[61] and Fiji[62].

## Statistics and reproducibility

Statistics were generated and calculated in GraphPad Prism 9. All data are presented as the mean ± standard deviation (S.D.) of at least three technical replicates unless otherwise stated. Statistical details (e.g. P-values, sample sizes, analysis type) for individual experiments are listed in Figure legends. Not all statistical comparisons are shown.

## Reporting summary

Further information on research design is available in the Nature Portfolio Reporting Summary linked to this article.

## Data availability

All data supporting the findings of this study are found within the paper and the Supplementary Information, or the Source Data. The materials generated in this study are available from corresponding authors under a standard materials transfer agreement between institutions. Source data are provided with this paper.

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

## Acknowledgements

This work was supported by NIH grant R01 AI114730 (to JCP), the Kwang Hua Educational Foundation (to JCP), by the NIAID-funded Center for Research on Influenza Pathogenesis and Transmission (CRIPT, Contract Number 75N93021C00014; to YK), and Grant 082098 from The Wellcome Trust (to SMH). We thank Ruben Donis of the Centers for Disease Control for providing membranes from the allantois and amnion for glycome analysis. We gratefully acknowledge the Centers for Disease Control and Prevention and International Reagent Resource (IRR; www. internationalreagentresource.org) for seasonal H3N2 virus samples utilized in this study.

## Author contributions

C.K., A.J.T., S.M.H., and J.C.P. conceived the project. A.J.T and C.K. cloned the plasmids. C.K. designed and conducted cell line development, HA protein purification, flow cytometry assays, first-round infectivity assays, and plaque size quantification. S.W. synthesized biotinylated glycans and performed glycan ELISA. A.A., R.K., and C.L. performed the N-glycan MS analysis. C.K. and T.M. grew and characterized virus stocks. T.M. performed virus replication assay. J.C.P., A.J.T., S.M.H., A.D., and Y.K. supervised the research. C.K., A.A., and J.C.P. wrote the initial draft of the manuscript and all authors assisted with editing.

## Competing interests

Y.K. has received collaborative research funds from FUJIFILM Toyama Chemical Co. LTD, Shionogi & Co. LTD, Daiichi Sankyo Pharmaceutical, Otsuka Pharmaceutical, KM Biologics, Kyoritsu Seiyaku, Fuji Rebio, Tauns Laboratories, Inc., Matsubara Co. LTD and is a cofounder of Flu-Gen. The other authors declare no competing interest with this work.
