## [Peer Review File · Nature Communications]

Glyco-engineered MDCK cells display preferred receptors of H3N2 influenza absent in eggs used for vaccinesREVIEWER COMMENTS

Reviewer #1 (Remarks to the Author):

Kikuchi et al. constructed knock-in cell lines (NExt lines) for the B3GNT2 gene in MDCK, SIAT and hCK cells with the aim to enhance synthesis of LacNAc repeats on glycans, and consequently, providing longer glycan antennae. The importance of this is in the fact that recent H3N2 virus isolates require 2-6 linked sialic acids on long antennae for binding. Thus, MDCK-based cell lines making longer antennae could be ideal for virus production.

The bulk of the paper is dealing with extensive glycomic analysis of the three parental and three knock-in cell lines. Glycomics was restricted to N-glycans, performed well, but presented in a difficult to access way. The main conclusion is that especially SIATNExt cells present a shift to larger N-glycans with longer chains. This correlates to enhanced binding and infection by recent H3N2 isolates.

The results are highly relevant in respect to a better understanding of the molecular details that make a SIATNExt cell line the cell line of choice for propagating recent H3N2 viruses in the lab. This is of relevance to fundamental research as well as to applications in the field of vaccine development. The obtained data and methodology are sound and support the conclusions. Still, a major revision is required to make the data accessible, and the text readable, for readers outside this highly specialized field of research.

Major comments.

1. Glycomics procedures are well described and performed but presentation is cumbersome. Whereas the large datasets that are produced can be analyzed (semi)quantitatively, this is only occasionally shown (Fig 3i). The text is overloaded with statements like “weakest”, “strongest”, “weak”, “strong”, “mostly”, “less abundant”, “very low relative”, “high occupancy”, “very small shift”, “very minor”, “substantially less”.....and so on. It makes reading a struggle with dozens of mass spec profiles to be compared. Tables (supplementary) are required where every peak is related to a single fixed standard (e.g. highest peak or total of all peaks). The most important data, as discussed in the text, could be supported by data extracted from these tables and presented in accessible way. Ample examples can be found in the literature.

2. In table 3, data are compared relative to a “base peak”. I agree that this way of presenting clearly displays a shift to higher complexity of glycans in cases (as shown by the colored “waves” in Fig. 3). However, the choice of the base peak is arbitrary as in principle any glycan with a single LacNAc repeat can be extended. Thereby, the choice creates an internal bias as the chosen peaks vary in relative abundance in between the different samples. Whereas determination of a ratio is still informative, calculating the % difference between conditions will give false results by the internal bias. It will be necessary to compare more absolute numbers (peak versus “total of peaks” for instance).

3. All binding data are performed with antibodies complexed HA to enable some sort of multivalent binding. This expectedly gives quite a different relative orientation/spacing of receptor binding sites

than on a virus particle. At least some of the data should be verified by comparing with a virus binding assay to correlate these results to the later infection experiments.

Other comments.

4. Line 55/56: Analysis was not done by SPR but by BLI.

5. Fig. 1b and methods: The scale of the figure is not visible. Quantification of the FACS data has not been well described in the methods section. For the MDCK cells a clear shift to higher binding is observed which does not seem to match with panel 2c for the same virus (Vic/11).

6. Fig. 2 Showing the FACS profiles as a supplementary data sets is needed.

7. Line 146, a 5-20 fold increase was suggested. Looking at the figure this looks more like 3-10 fold. A Supp table with the data would be useful.

8. Line 167-169. It is stated that complex N-glycans were mostly capped with NeuAc. But actually the glycans of panel 2 supp fig. 1, which are abundant, are mostly uncapped.

9. HA binding was compared to entry efficiency in a single round infection assay (lines 305-321). However the assay conditions for binding and entry differ a lot. HA binding was performed for 2h at room temperature (line 497) and readout can be considered as an endpoint at which binding has reached equilibrium. In single round entry experiments virus was incubated at a much higher temperature of 33oC for only 10 minutes (line 568-569). In addition, another set of virus strains was used for entry than for binding (except for WI/18). At least HK/68 and Camb/20 should be include in the infection assay. In addition, it should tested whether a longer incubation of virus with cells in the entry assay will reduce the difference in entry efficiency for the different cell lines. In this way virus binding will be closer to equilibrium/endpoint binding levels. This may also reduce the observed difference between single round infection Fig 5a) and virus titration (Fig 5b).

10. The discussion section is to a large extent a summary of the results without adding a lot of discussion. Especially lines 371 to 390 are mostly repeat.

11. In the discussion reference is made to two papers (references 42 and 25) that describe a (more limited) glycomic analysis of part of the cell lines analyzed here. These studies describe remarkable differences with the current study like extensive sulfation or the abundant presence of bi-secting GlcNAc. These differences are ignored here but are essential to be discussed for a more comprehensive understanding of the complexities involved in changing glycan composition by knock-in of glycosyl transferases.

12. Although a shift to higher MW was convincingly shown, the authors should discuss that in absolute quantities this shift is still very small. The absolute density of sialylated glycans with extended chain length will still be very low. Given the requirement for multivalent binding, as affinity HA for sialic acid is very low, this issue requires discussion.

Reviewer #2 (Remarks to the Author):

The manuscript describes an attempt to make a better cell line for recent H3N2 viruses, which have receptor binding specificity toward extended glycan receptor with multiple internal LacNAc repeats. This was done by overexpressing an enzyme supposedly responsible for synthesis of this glycan type. The new cell lines showed enhanced binding and susceptibility to recent H3N2 isolates and more abundance of the predicted glycan. However, the cell lines did not improve the final yield of the viruses due to a delay in multiple-cycle viral kinetics likely due to reduced viral release because of the enhanced binding. This makes the cell lines less useful for vaccine production as intended. Here are my specific comments:

1. Only one clone with the highest binding to H3 HA was shown for one cell line in all the experiment. Were there much diversity in phenotype among clones from the same parental cell line? Is it possible that choosing the clones with highest binding made them less productive in viral release because of too strong binding resulting in overall unimproved yield? Did the authors test clones with moderately improved binding for viral production?
2. The SIAT-NExt cell line may be still useful for viral isolation as it showed enhanced susceptibility, which should improve sensitivity and isolation rate. More discussion on this point may be added.
3. The reason for the changed receptor specificity is the presence of these glycan in human airway. How are these cell lines compared to human airway epithelial cell lines in term of H3N2 binding and susceptibility?
4. This receptor specificity to the extended glycan is found not only in H3 but also in H1pdm. More explanation may be added and the reference cited at the bottom of Page 6 may be incorrect. Should it be ref #22 instead of #30?

Prasert Auewarakul

Reviewer #3 (Remarks to the Author):

Evolution of human H3N2 influenza viruses driven by immune selection has narrowed the receptor specificity of the hemagglutinin (HA) to a restricted subset of human-type (Neu5Aca2-6Gal) glycan receptors that have extended poly-LacNAc (Gal β 1-4GlcNAc) repeats. To assess the impact of extended glycan receptors on virus binding, infection, and growth, the authors have engineered N-glycan extended (NExt) cell lines by overexpressing β 3-N-acetylglucosaminyltransferase 2 (β 3GnT2) in MDCK, SIAT, and hCK cell lines. Of these, SIAT-NExt cells exhibited markedly increased binding of H3 HAs and susceptibility to infection by recent H3N2 virus strains. Glycome analysis of these cell lines and allantoic

and amniotic egg membranes provide insights into the importance of extended glycan receptors for growth of recent H3N2 viruses and relevance to their production for cell- and egg-based vaccines.

The presentation reflects the present state of knowledge and the data are attained from a large amount of literature and the author's group supported by the line of reasoning. The description in the text is very clear and easy to follow. The graphical presentation is truthful and easy to understand. The table present data is also clear and truthful. I recommend to consideration for the publication of this article. However, the authors need to address some concerns prior to its publication, see below for details.

(1) Three parental lines and their NExt counterparts were characterized for expression of α 2-3 and α 2-6 sialosides by flow cytometry using SNA, a lectin specific for NeuAca2-6Gal, and an H5 rHA specific for NeuAca2-3Gal from an avian virus A/Viet Nam/1203/2004 (H5N1; Viet/04) (Fig. 2a). Here, why do the authors not use a lectin specific for NeuAca2-3Gal, It is more accurate? Maybe, this H5 rHA also can partly bind NeuAca2-6Gal?

(2) Line 145-147, overexpression of the β 3GnT2 enzyme resulted in 5-20 fold increased binding of the HAs to SIAT-NExt cells relative to SIAT cells, and 1.5-3 fold increased binding to hCK-NExt cells relative to hCK cells. Here, the authors should provide the evidence.

(3) This is a problem that is worth to discuss. Human H3N2 viruses isolated after 2000 have also been widely reported to be more difficult to grow in Madin-Darby canine kidney (MDCK) cells. Evolution of human H3N2 influenza viruses driven by immune selection has narrowed the receptor specificity of the HA to a restricted subset of human-type (Neu5Aca2-6Gal) glycan receptors that have extended poly-LacNAc (Gal β 1-4GlcNAc) repeats. if the altered glycosylation of the HA in human H3N2 virus strains resulted in this narrowed receptor specificity?

(4) Comparing to MDCK, SIAT, the full name of "hCK" should be mentioned in the manuscript. Meanwhile, it lacks of the description of glycobiological characters. Similar, the full name of "MFI" in Line 117 is missing.

(5) "Glycomics profiles and Glycomics analysis" should change to "Glycome profiles and Glycome analysis"

Point-by-point response to the reviewers

REVIEWER COMMENT

Reviewer #1 (Remarks to the Author):

Kikuchi et al. constructed knock-in cell lines (NExt lines) for the B3GNT2 gene in MDCK, SIAT and hCK cells with the aim to enhance synthesis of LacNAc repeats on glycans, and consequently, providing longer glycan antennae. The importance of this is in the fact that recent H3N2 virus isolates require 2-6 linked sialic acids on long antennae for binding. Thus, MDCK-based cell lines making longer antennae could be ideal for virus production.

The bulk of the paper is dealing with extensive glycomics analysis of the three parental and three knock-in cell lines. Glycomics was restricted to N-glycans, performed well, but presented in a difficult to access way. The main conclusion is that especially SIATNExt cells present a shift to larger N-glycans with longer chains. This correlates to enhanced binding and infection by recent H3N2 isolates.

The results are highly relevant in respect to a better understanding of the molecular details that make a SIATNExt cell line the cell line of choice for propagating recent H3N2 viruses in the lab. This is of relevance to fundamental research as well as to applications in the field of vaccine development. The obtained data and methodology are sound and support the conclusions. Still, a major revision is required to make the data accessible, and the text readable, for readers outside this highly specialized field of research.

We thank the reviewer for the positive comments and constructive suggestions for improvement of the manuscript and accessibility of the data to the readership.

Major comments.

1. Glycomics procedures are well described and performed but presentation is cumbersome. Whereas the large datasets that are produced can be analyzed (semi)quantitatively, this is only occasionally shown (Fig 3i). The text is overloaded with statements like “weakest”, “strongest”, “weak”, “strong”, “mostly”, “less abundant”, “very low relative”, “high occupancy”, “very small shift”, “very minor”, “substantially less”.....and so on. It makes reading a struggle with dozens of mass spec profiles to be compared. Tables (supplementary) are required where every peak is related to a single fixed standard (e.g. highest peak or total of all peaks). The most important data, as discussed in the text, could be supported by data extracted from these tables and presented in accessible way. Ample examples can be found in the literature.

2. In table 3, data are compared relative to a “base peak”. I agree that this way of presenting clearly displays a shift to higher complexity of glycans in cases (as shown by the colored “waves” in Fig. 3). However, the choice of the base peak is arbitrary as in principle any glycan with a single LacNAc repeat can be extended. Thereby, the choice creates an internal bias as the chosen peaks vary in relative abundance in between the different samples. Whereas determination of a ratio is still informative, calculating the % difference between conditions will give false results by the internal bias. It will be necessary to compare more absolute numbers (peak versus “total of peaks” for instance).

We thank the reviewer for these important comments. In the revised manuscript we have substantively simplified the presentation of the glycomics analysis in the results and discussion to make the key observations more accessible to the reader. This is greatly aided by our response to the second point, namely, to present the glycomics data in a way that allows comparisons between cells in an unbiased way.

In the revised manuscript, the relative intensities of the molecular ions corresponding to complex N-glycans (m/z 2966 and above) were normalized to the sum of their relative intensities. This allows us to compare changes afforded by the β 3GnT2 in a more reliable manner, as we have done in Fig. 3e-i. In Fig. 3e-h we express these percentage changes of the SIAT vs SIAT-NExt and hCK vs hCK-NExt to visually show the impact of the expression of β 3GnT2, and show the same in Fig. 3i in tabular form. Of note, in the Supplementary Data 1, we also show the percentage change having as base peak the relative intensity corresponding to the molecular ion at m/z 2966 (for comparison, but not discussed in the text).

We now provide the data in this way for the spectra of all cell lines in Excel spreadsheet in Supplementary Data 1.

3. All binding data are performed with antibodies complexed HA to enable some sort of multivalent binding. This expectedly gives quite a different relative orientation/spacing of receptor binding sites than on a virus particle. At least some of the data should be verified by comparing with a virus binding assay to correlate these results to the later infection experiments.

As mentioned, we have historically used antibody complexes of the his-tagged HA using anti-His-tag and anti-anti-His-tag antibodies to create tetramers of the HA trimers to achieve sufficient avidity of the HAs for binding to glycan arrays, ELISA plates, and cell surfaces (Stevens, 2006). This is needed since the intrinsic affinity of the HAs for natural sialosides is \sim 1-2 mM. These complexes also amplify the differences in recognition of avian-type (α 2-3) and human-type (α 2-6) receptors to reflect the specificity exhibited by the dense expression of HAs on the surface of the virus. Similarly, these complexes are able to clearly show differences in binding to cells that differ in their expression of NeuAc α 2-6Gal extended glycan receptors.

We agree it is important to show that the virus exhibits similar binding characteristics as the recombinant HAs. For this reason, in the single-round infectivity assay (Fig. 5a and b in the revised manuscript), we use a short time for adsorption of viruses to cells (10 minutes at 33°C) to so that the infection reflects the efficiency/early kinetics of HA-mediated binding to cell surface. At longer incubation times binding is no longer the rate limiting step, and the differences are minimized. In this assay, since infectious virus is not produced (by not having TPCK trypsin that cleaves HA0 into HA1 and HA2), the number of infected cells visualized after 16 hours is a direct measure of how many cells bound and internalized virus.

As suggested, to provide a direct comparison with the HA binding assay we have added HK/68 and Vic/11 to the single-round infectivity assay to effectively to allow more direct comparison between the HA binding assays and the single-round infectivity assay when the binding step is rate limiting. We have accordingly revised the manuscript to point out the comparison between the two assays (Fig 5a).

Other comments.

4. Line 55/56: Analysis was not done by SPR but by BLI.

Corrected, thank you.

5. Fig. 1b and methods: The scale of the figure is not visible. Quantification of the FACS data has not been well described in the methods section. For the MDCK cells a clear shift to higher binding is observed which does not seem to match with panel 2c for the same virus (Vic/11).

Thank you for these suggestions:

- Values for the x-axis have now been added and font size increased.
- The quantification of the flow cytometry data and how ratios from the flow data were calculated for Fig. 2c is now better described in methods.
- The clear shift seen for higher binding for MDCK cells in Fig. 1b is also seen as a statistically significant difference in Fig. 2c (Vic/11 panel).

6. Fig. 2 Showing the FACS profiles as a supplementary data sets is needed.

We added Supplementary Figure 2 with flow diagrams representative for each agent in Fig. 2a-c.

7. Line 146, a 5-20 fold increase was suggested. Looking at the figure this looks more like 3-10 fold. A Supp table with the data would be useful.

Thank you for pointing this out. The fold change mentioned in the text was relative to the background (dotted line). We have now cited the fold change comparing the parental cells to the respective β 3GNT2 transfected cells, with up to 10 fold change for SIAT vs SIAT-NExt.

8. Line 167-169. It is stated that complex N-glycans were mostly capped with NeuAc. But actually the glycans of panel 2 supp fig. 1, which are abundant, are mostly uncapped.

Supplementary Figure 1 is now Supplementary Figure 3. The molecular ion peaks with α 2-3-linked sialic acids are highlighted in red in Sup. Fig. 3a (e.g. panel 1) as stated in the figure legend. The same peaks are seen in Sup. Fig. 3b (panel 2), but they are not highlighted in red. The fact that they are not highlighted may have been the reason they were assumed to be uncapped. In contrast, most of these peaks are diminished or absent in the panels showing glycans treated with sialidase Sup. Figs 3c and 3d. These differences are highlighted in green in Fig. 3c to show which peaks are increased as a result of sialidase treatment. The use of these colors in Fig. 3a and Fig. 3c was to highlight the differences before and after sialidase treatment.

9. HA binding was compared to entry efficiency in a single round infection assay (lines 305-321). (a) However the assay conditions for binding and entry differ a lot. HA binding was performed for 2h at room temperature (line 497) and readout can be considered as an endpoint at which binding has reached equilibrium. In single round entry experiments virus was incubated at a much higher temperature of 33oC for only 10 minutes (line 568-569). (b) In addition, another set of virus strains was used for entry than for binding (except for WI/18). At least HK/68 and Camb/20 should be include in the infection assay. (c) In addition, it should tested whether a longer incubation of virus with cells in the entry assay will reduce the difference in entry efficiency for the different cell lines. In this way virus binding will be closer to equilibrium/endpoint binding levels. (d) This may also reduce the observed difference between single round infection Fig 5a) and virus titration (Fig 5b).

We inserted (a) to (d) because this comment has multiple subpoints.

(a) We agree this is an important point, and we chose to address it in comment 3 when it was first raised. We do not consider either assay as equilibrium binding, particularly in such a multivalent context where we are trying to demonstrate observable binding, particularly in the HA binding assay. We submit that the short incubation time for the cell

infection experiment is key to assess the relative ability of the virus to adsorb to the cell prior to internalization and infection.

(b) We agree and have added HK/68 and Vic/11 (Camb/20 virus was not available) to enable comparison between HA-avidity assay and single-round infectivity assay (Fig. 5a in revised manuscript).

(c) Our pilot experiments were done with 30 min adsorption, and under these conditions there were smaller differences, so we used 10 min adsorption.

(d) We believe there are major differences between the single-round infection assay (Fig. 5a,b) and the kinetics of multi-round virus replication (not titration) assay (Fig. 5c). The single round assay helps 'isolate' the adsorption step mediated by the HA. The multi-round virus replication assay includes virus adsorption, virus internalization mediated by the HA, and release from the cell to infect other cells, which involves both the HA and the NA that cleaves sialic-acid containing receptor. In this regard, we speculate that higher avidity of the HA for extended glycans and resulting insufficient NA activity is the basis for slower kinetics of replication of some of the viruses, and the reduced size of the plaques for WI/18 on SIAT-NExt cells (Fig. 5d in the revised manuscript). These results depict the critical aspect of virus biology that we were not able to find with assays focusing on HA avidity alone, and indeed an interesting point to discuss.

10. The discussion section is to a large extent a summary of the results without adding a lot of discussion. Especially lines 371 to 390 are mostly repeat.

We agree and have revised the discussion to reduce redundancy and add discussion points raised by the three reviewers.

11. In the discussion reference is made to two papers (references 42 and 25) that describe a (more limited) glycomic analysis of part of the cell lines analyzed here. These studies describe remarkable differences with the current study like extensive sulfation or the abundant presence of bi-secting GlcNAc. These differences are ignored here but are essential to be discussed for a more comprehensive understanding of the complexities involved in changing glycan composition by knock-in of glycosyl transferases.

Thank you for the suggestion. We revised the results section and discussion section concerning this point.

12. Although a shift to higher MW was convincingly shown, the authors should discuss that in absolute quantities this shift is still very small. The absolute density of sialylated glycans with extended chain length will still be very low. Given the requirement for multivalent binding, as affinity HA for sialic acid is very low, this issue requires discussion.

In the revised Fig. 3 e-h, we show that there is a major increase in the extended glycans in SIAT-NExt relative to SIAT cells, while the increase in hCK-NExt relative to hCK is much less. However, since our data do not allow us to quantitatively compare the absolute quantities of extended branches on SIAT-NExt vs hCK-NExt, we are not able to definitively document the precise reason SIAT-NExt exhibits better binding of H3 HA and H3N2 virus. In the discussion we have now included a new paragraph to discuss this limitation and possible variation of structural isomer that could have influenced the biological outcome.

Reviewer #2 (Remarks to the Author):

The manuscript describes an attempt to make a better cell line for recent H3N2 viruses, which

have receptor binding specificity toward extended glycan receptor with multiple internal LacNAc repeats. This was done by overexpressing an enzyme supposedly responsible for synthesis of this glycan type. The new cell lines showed enhanced binding and susceptibility to recent H3N2 isolates and more abundance of the predicted glycan. However, the cell lines did not improve the final yield of the viruses due to a delay in multiple-cycle viral kinetics likely due to reduced viral release because of the enhanced binding. This makes the cell lines less useful for vaccine production as intended.

We thank specific comments that encouraged us to further strengthen our manuscript and discussion.

Here are my specific comments:

1. Only one clone with the highest binding to H3 HA was shown for one cell line in all the experiment. **(a)** Were there much diversity in phenotype among clones from the same parental cell line? **(b)** Is it possible that choosing the clones with highest binding made them less productive in viral release because of too strong binding resulting in overall unimproved yield? Did the authors test clones with moderately improved binding for viral production?

We have inserted (a) and (b) to address each point separately.

(a) There was variation in Vic/11 rHA binding among the single cell clones. We have added Supplementary Figure 1 that show flow cytometry data for HA binding to the top 3 transfected clones for each parental cell lines, which was the basis for choosing the best clone for each line.

(b) While we selected the ‘best’ of three β 3GNT2 transfected clones from each parental line (MDCK, SIAT, and hCK) based on binding of Vic/11 rHA, the three clones differed little from each other (see new Supplementary Fig. 2), so we have not tried to look for differences in multiple clones from the same parental line.

That said, we agree with the point made by the reviewer that increased avidity could improve virus adsorption but could impact virus replication due to impaired release from the infected cell. Indeed, we believe that this may be the reason for the reduced kinetics of viral replication for some H3N2 viruses (Fig. 5c) and smaller plaque size for W1/18 virus on SIAT-NExt cells (Fig. 5d). We have now commented on this in the discussion.

2. *The SIAT-NExt cell line may be still useful for viral isolation as it showed enhanced susceptibility, which should improve sensitivity and isolation rate. More discussion on this point may be added.*

Thank you for emphasizing this important point, and we have mentioned it in the discussion.

3. *The reason for the changed receptor specificity is the presence of these glycan in human airway. How are these cell lines compared to human airway epithelial cell lines in term of H3N2 binding and susceptibility?*

We have been working on this exact point for nearly 5 years in collaboration with groups at Northwestern University and Imperial College London. We have obtained human nasal and tracheal primary epithelial cells samples and conducted detailed glycomics on both the primary cells and the donor-matching air-liquid-interface (ALI) cultured cells. We do indeed find that extended glycans are present on the human airway primary epithelial cells. We have also found that ALI-cultured cells undergo a glycan remodeling due to altered expression pattern of glycosyltransferases. For this reason, it is not clear that epithelial cell lines or even ALI-cultured primary cells represent a more physiologically

relevant model than the MDCK cell lines described here. A manuscript summarizing this work is in preparation.

4. This receptor specificity to the extended glycan is found not only in H3 but also in H1pdm. More explanation may be added and the reference cited at the bottom of Page 6 may be incorrect. Should it be ref #22 instead of #30?

Thank you for pointing this out. Indeed it was an incorrect citing, we have now corrected the citations (ref # 35 and 36.) We agree that H1N1 has preference for extended glycans, but for two or more LacNAc units instead of three or more for H3N2. This difference is evident in the binding of the an exemplary H1 HA (Hawaii/19) in Fig. 2b, where binding to hCK and hCK-NExt cells is preferred. This is now more clearly discussed in the results section.

Reviewer #3 (Remarks to the Author):

Evolution of human H3N2 influenza viruses driven by immune selection has narrowed the receptor specificity of the hemagglutinin (HA) to a restricted subset of human-type (Neu5Aca2-6Gal) glycan receptors that have extended poly-LacNAc (Gal β 1-4GlcNAc) repeats. To assess the impact of extended glycan receptors on virus binding, infection, and growth, the authors have engineered N-glycan extended (NExt) cell lines by overexpressing β 3-N-acetylglucosaminyltransferase 2 (β 3GnT2) in MDCK, SIAT, and hCK cell lines. Of these, SIAT-NExt cells exhibited markedly increased binding of H3 HAs and susceptibility to infection by recent H3N2 virus strains. Glycome analysis of these cell lines and allantoic and amniotic egg membranes provide insights into the importance of extended glycan receptors for growth of recent H3N2 viruses and relevance to their production for cell- and egg-based vaccines. The presentation reflects the present state of knowledge and the data are attained from a large amount of literature and the author's group supported by the line of reasoning. The description in the text is very clear and easy to follow. The graphical presentation is truthful and easy to understand. The table present data is also clear and truthful. I recommend to consideration for the publication of this article. However, the authors need to address some concerns prior to its publication, see below for details.

We thank the reviewer for positive review and comments for improvement of the manuscript.

(1) Three parental lines and their NExt counterparts were characterized for expression of a2-3 and a2-6 sialosides by flow cytometry using SNA, a lectin specific for NeuAca2-6Gal, and an H5 rHA specific for NeuAca2-3Gal from an avian virus A/Viet Nam/1203/2004 (H5N1; Viet/04) (Fig. 2a). Here, why do the authors not use a lectin specific for NeuAca2-3Gal, It is more accurate? Maybe, this H5 rHA also can partly bind NeuAca2-6Gal?

MAA (*Maackia Amurensis*) lectins (I or II) are often utilized for the detection of NeuAca2-3Gal. However, detailed analysis of the specificity of these lectins (www.functionalglycomics.org) show that they have strongest specificity for sulfated-LacNAc glycans followed by NeuAca2-3Gal glycans, and depending on supplier, also bind to non-sialylated poly-LacNAc glycans and other non-sialylated glycans. We have instead used Viet/04 rHA for the detection of NeuAca2-3Gal terminal structure because we have found this is one of the most reliable probes for this structure. We have previously reported the strict specificity of this HA for glycans containing the NeuAca2-3Gal epitope (Citation# 34, Thompson et. al. Cell Host & Microbe (2020) Fig.2C with mAb detection).

(2) Line 145-147, overexpression of the β 3GnT2 enzyme resulted in 5-20 fold increased binding of the HAs to SIAT-NExt cells relative to SIAT cells, and 1.5-3 fold increased binding to hCK-NExt cells relative to hCK cells. Here, the authors should provide the evidence.

Thank you for pointing this out as also mentioned by reviewer 1. We mistakenly used the fold difference relative to the background. We now report the fold change for the comparison between SIAT and SIAT-NExt, or hCK and hCK-NExt. (Lines 142-144)

(3) This is a problem that is worth to discuss. Human H3N2 viruses isolated after 2000 have also been widely reported to be more difficult to grow in Madin-Darby canine kidney (MDCK) cells. Evolution of human H3N2 influenza viruses driven by immune selection has narrowed the receptor specificity of the HA to a restricted subset of human-type (Neu5Aca2-6Gal) glycan receptors that have extended poly-LacNAc (Gal β 1-4GlcNAc) repeats. Has the altered glycosylation of the HA in human H3N2 virus strains resulted in this narrowed receptor specificity?

Indeed, the number of N-linked glycans on the HA has increased from 6 in the pandemic virus, to 11-12 now. We previously addressed this question when we first reported the evolved specificity for extended glycans (Citation # 22, Peng et. al. *Cell Host Microbe* (2017), Fig 6). We produced the Vic/11 HA, which has an extended glycan specificity, with high mannose N-glycans in HEK293S cells. The glycans were then quantitatively removed with Endo H, leaving a single sugar (GlcNAc-Asn) stub. We found that the specificity of the HA for extended glycans was identical to the fully glycosylated HA produced in HEK293 cells. We concluded that N-glycosylation of HA does not alter the glycan recognition nor receptor specificity.

(4) Comparing to MDCK, SIAT, the full name of "hCK" should be mentioned in the manuscript. Meanwhile, it lacks of the description of glycobiological characters. Similar, the full name of "MFI" in Line 117 is missing.

Thank you for pointing these out. We added the full name to define these abbreviations.

(5) "Glycomics profiles and Glycomics analysis" should change to "Glycome profiles and Glycome analysis"

The changes have been made as suggested.

REVIEWERS' COMMENTS

Reviewer #1 (Remarks to the Author):

The authors made a large effort on presenting the quantitative glycomics in a clear way. Especially fig 3 and the supplementary tables are very useful. Most comments were answered adequately. I still have some doubts concerning the argument that binding of recombinant HA can be related to binding of virus particles with infection as a read out. But I understand that further experiments on this matter are beyond the scope of this paper.

Reviewer #2 (Remarks to the Author):

The authors have satisfactorily responded to all my comments. I have no further comments.

Reviewer #3 (Remarks to the Author):

The authors revised the manuscript according to suggestions of the reviewers, no further comments.

Point-by-point response to the reviewers

REVIEWER COMMENT

Reviewer #1 (Remarks to the Author):

The authors made a large effort on presenting the quantitative glycomics in a clear way. Especially fig 3 and the supplementary tables are very useful. Most comments were answered adequately. I still have some doubts concerning the argument that binding of recombinant HA can be related to binding of virus particles with infection as a read out. But I understand that further experiments on this matter are beyond the scope of this paper.

We thank the reviewer for the constructive suggestions in the previous review cycle to improve the presentation of our quantitative glycomics data and our manuscript.

Reviewer #2 (Remarks to the Author):

The authors have satisfactorily responded to all my comments. I have no further comments.

Reviewer #3 (Remarks to the Author):

The authors revised the manuscript according to suggestions of the reviewers, no further comments.

We thank the reviewers for the excellent comments and suggestions in the previous review cycle to improve our manuscript.